# MUTUAL-INFORM SMOE: IMPROVING ROUTING STABILITY VIA PROBABILISTIC GRAPHICAL MODEL

## ABSTRACT

Sparse Mixture of Experts (SMoE) has emerged as a breakthrough approach for achieving unprecedented scalability in deep learning. By enabling models to expand their parameter count exponentially while selectively activating only a small subset of parameters per sample, SMoEs maintain high efficiency. However, SMoE models are susceptible to routing fluctuations, leading to instability and non-robustness. In this work, we unveils SMoE-based attention as a point estimate of a regression function of a three-layer hierarchical mixture of experts regression. Through this probabilistic graphical model (PGM) framework, we highlight the conditional independence in expert-selection process of tokens, which exposes the model to routing fluctuation and non-robustness. Motivated by this PGM framework, we propose Mutual-Inform SMoEs, including Similarity and Attention-Inform SMoE, which eliminate the assumption of conditional independence by allowing tokens to directly influence each other on expert-decisions. We theoretically demonstrate that our methods lower the entropy in decision-making, enabling more confident and consistent expert assignments. Finally, we empirically validate our models on ImageNet classification and Wikitext-103 language modeling, showing significant improvements in reducing routing fluctuations, enhancing performance, and increasing model robustness compared to baseline Transformer-SMoE models.

## 1 INTRODUCTION

Mixture of Experts (MoEs) (Jacobs et al., 1991; Jordan & Jacobs, 1994) has been widely used as a principle approach to scale up the number of parameters of deep neural networks while introducing an affordable computation. As a result, MoEs appears in almost all applications of machine learning and deep learning including large language model (Devlin et al., 2018; Radford et al., 2019; Raffel et al., 2020; Kaplan et al., 2020; Brown et al., 2020; Touvron et al., 2023), vision understanding (Neil & Dirk, 2020; Bao et al., 2021; 2022; Li et al., 2023; Bai et al., 2024), and many other applications (Subramanian et al., 2024; Gaur et al., 2021; Gormley & Murphy, 2011). A recent variation of Mixture of Experts (MoEs), called Sparse MoEs (SMoEs) (Shazeer et al., 2017), has been introduced to enhance model size to billion-parameter while maintaining constant computational costs by modularizing the network and activating only specific subsets of experts for each input. Therefore, SMoEs has been applied successfully in translation models (Lepikhin et al., 2020), pre-training (Fedus et al., 2022; Artetxe et al., 2021), GPT-3 level one-shot performance (Du et al., 2022), image classification (Riquelme et al., 2021), and so on.

### 1.1 BACKGROUND MULTIHEAD ATTENTION

For a given input sequence $\mathbf{X} := [\boldsymbol{x}_1, \cdots, \boldsymbol{x}_N]^\top \in \mathbb{R}^{N \times D_x}$ of $N > 1$ feature vectors in $D_x \geq 1$ dimensions, self-attention transforms $\mathbf{X}$ into the output sequence $\mathbf{H}$ in the following two steps:

**Step 1.** Given each attention head $h$, the input sequence $\mathbf{X}$ is projected into the query matrix $\mathbf{Q}_h$, the key matrix $\mathbf{K}_h$, and the value matrix $\mathbf{V}_h$ via three linear transformations: $\mathbf{Q}_h = \mathbf{X}\mathbf{W}_{Q,h}^\top; \mathbf{K}_h = \mathbf{X}\mathbf{W}_{K,h}^\top; \mathbf{V}_h = \mathbf{X}\mathbf{W}_{V,h}^\top$, where $\mathbf{W}_{Q,h}, \mathbf{W}_{K,h} \in \mathbb{R}^{D \times D_x}$, and $\mathbf{W}_{V,h} \in \mathbb{R}^{D_v \times D_x}$ ($D \geq 1$) are the weight matrices. We denote $\boldsymbol{Q}_h := [\boldsymbol{q}_{1,h}, \cdots, \boldsymbol{q}_{N,h}]^\top, \mathbf{K}_h := [\boldsymbol{k}_{1,h}, \cdots, \boldsymbol{k}_{N,h}]^\top$, and $\mathbf{V}_h := [\boldsymbol{v}_{1,h}, \cdots, \boldsymbol{v}_{N,h}]^\top$, where the vectors $\boldsymbol{q}_{i,h}, \boldsymbol{k}_{i,h}, \boldsymbol{v}_{i,h}$ for $i = 1, \cdots, N$ are the query, key, and value vectors, respectively.

**Step 2.** The output sequence is then computed as $\mathbf{H}_h = \mathrm{softmax}\left(\mathbf{Q}_h \mathbf{K}_h^\top / \sqrt{D}\right)\mathbf{V_h} := \mathbf{A}_h \mathbf{V}_h$, where the softmax function is applied to each row of the matrix $\mathbf{A} = \mathrm{softmax}(\mathbf{Q}_h \mathbf{K}_h^\top)$. This matrix $\mathbf{A}_h \in \mathbb{R}^{N \times N}$ and its component $a_{ij,h}$ for $i,\ j = 1, \cdots, N$ are called the attention matrix and attention scores for head $h$, respectively.

**Multi-head Attention (MHA)** In MHA, multiple heads are concatenated to compute the final output. Let $H \geq 1$ be the number of heads and $\mathbf{W}'^O \in \mathbb{R}^{HD \times HD}$ be the projection matrix for the output. The multi-head attention is defined as

$$\bar{\mathbf{U}} = \mathrm{MHA}(\mathbf{X}) = \mathrm{Concat}(\mathbf{H}_1, \ldots, \mathbf{H}_H)\mathbf{W}'_O = \frac{1}{H}\sum_{h=1}^{H} \mathbf{A}_h \mathbf{V}_h \mathbf{W}_{O,h}, \tag{1}$$

where $[\mathbf{W}_{O,1}, \ldots, \mathbf{W}_{O,H}] = H\mathbf{W}'_O$ and $\mathbf{W}_{O,h} \in \mathbb{R}^{D \times HD}$.

## 1.2 Background Sparse Mixture of expert in Transformer

A (Sparse) Mixture-of-Experts ((S)MoE) model consists of a router and $K$ experts. For each input token $\bar{\mathbf{u}}_i \in \mathbb{R}^D$, the router computes the affinity scores between $\bar{\mathbf{u}}_i$ and each expert as $r_k(\bar{\mathbf{u}}_i)$, where $k = 1, 2, \ldots, K$. The router's score $\mathbf{r}(\bar{\mathbf{u}}_i) = [r_1(\bar{\mathbf{u}}_i), r_2(\bar{\mathbf{u}}_i), \ldots, r_K(\bar{\mathbf{u}}_i)]^\top = \mathbf{W}\bar{\mathbf{u}}_i + \boldsymbol{b}$, where $\mathbf{W} \in \mathbb{R}^{K \times D}$ and $\boldsymbol{b} \in \mathbb{R}^K$. MoE then takes the softmax of the router scores as coefficients for a linear combination of expert outputs $\boldsymbol{g}_k(\bar{\mathbf{u}}_i)$. To increase the capacity of the Transformer model while not incur a heavy additional computation, SMoE instead uses a sparse gating function as a router, selecting only $M$ experts with the highest affinity scores. The TopM function is defined as:

$$\mathrm{TopM}(\mathbf{r})[k] := \begin{cases} r_k, & \text{if } r_k \text{ is among the } M \text{ largest elements of } \mathbf{r} \\ -\infty, & \text{otherwise} \end{cases}$$

The outputs from the $M$ selected experts are then linearly combined as:

$$\bar{\mathbf{o}}_i = \sum_{k=1}^{K} \mathrm{softmax}(\mathrm{TopM}(\mathbf{r}(\bar{\mathbf{u}}_i))[k])\mathbf{g}_k(\bar{\mathbf{u}}_i) \tag{2}$$

We discuss the renormalization in Section E.

**Routing fluctuation in SMoE.** One of the major challenges in training SMoE Transformers is the instability caused by fluctuating routing decisions during training (Dai et al., 2022; Zoph et al., 2022; Chi et al., 2022). This instability leads to model non-robustness. Improving the consistency of expert routing decisions is critical for model stability and overall model performance, since routing fluctuations, especially in the later stages of training make it challenging to determine an appropriate stopping point for training. For instance, even in the final epochs of training, upto 33% of tokens still switch their assigned experts (c.f. Figure 2). This can result in different behavior during inference depending on when training is halted. Therefore, reinforcing consistent routing decisions enhances model robustness and improves overall performance.

## 1.3 Contributions

We develop a probabilistic graphical model (PGM) framework for the attention-(S)MoE block, within which we highlight the conditional independence in expert selection for each individual token, a property that makes it prone to routing fluctuations. Building on this insight, we propose a novel notion of (S)MoE, named Mutual-Inform (S)MoE, which encourages the assignment of similar tokens to the same expert. By letting other tokens directly influence one's routing decision, we demonstrate that the model reduces routing fluctuations, resulting in enhanced robustness.

Our contributions are four-fold:

- We present a novel probabilistic graphical framework (PGM) revealing that attention-(S)MoE is a point estimate of the regression function in a three-layer hierarchical mixture of expert regression. Through this PGM perspective, we show the conditional independence in token's expert selection, leading to routing fluctuation, and model non-robustness.

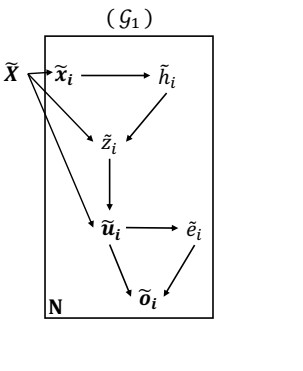
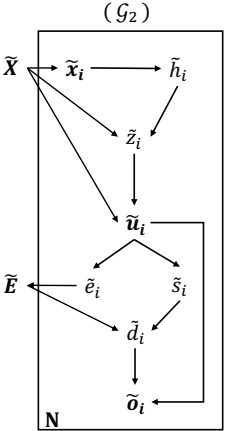
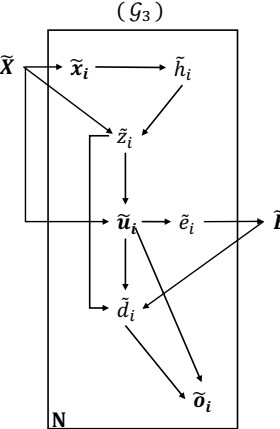

Figure 1: PGM for Baseline-MoE ($\mathcal{G}_1$), Attention-Inform MoE ($\mathcal{G}_2$), Similarity-Inform MoE ($\mathcal{G}_3$). In this figure, $\tilde{\mathbf{X}}$ denotes the random variable of the input sequence, $\tilde{\mathbf{x}}_i$ is the $i$-th token in the $X$ sequence, $\tilde{h}_i$ is the index of the selected head for token $i$-th, $\tilde{z}_i$ denotes the index of the attention head and position, and $\tilde{\mathbf{E}}$ denotes the stacked matrix of $\tilde{e}_1, \ldots, \tilde{e}_N$

- Within the PGM framework, we propose a novel notion of (Sparse) Mixture-of-Experts, named **Mutual-Inform (S)MoE**, where expert decisions are no longer made conditionally independently from each token. Instead, tokens influence each other's expert selection based on their similarities and relevance. This correspondence can be computed directly between the input tokens in the MoE layers, or derived from the attention layer, yielding two mechanism of Mutual-Inform (S)MoE: Similarity-Inform (S)MoE and Attention-Inform (S)MoE.

- In our theoretical analysis, we show that our methods reduce the entropy in decision-making processes of indecisive tokens. This reduction in entropy facilitates more confident and consistent expert assignments.

- Finally, we demonstrate the advantages and robustness of Mutual-Inform SMoE models across various tasks and domains, including ImageNet classification and Wikitext-103 language modeling.

**Organisation.** We first interpret the attention-(S)MoE through the lens of probabilistic graphical model in Section 2. Leveraging this new insight, we discuss novel class of (S)MoEs, named Mutual-Inform (S)MoE in Section 3. Experiments on language modeling, ImageNet classification, and empirical analysis are given in Section 4. Finally, we conclude the paper in Section 6. Additional materials are deferred to the Appendices.

**Notation.** We denote a random matrix as $\tilde{\mathbf{X}}$, and its specific realization as $\mathbf{X}$. Similarly, a random vector is represented by $\tilde{\mathbf{x}}$, with its realization as $\mathbf{x}$. Scalars random variable are are denoted by non-bold letters, such as $\tilde{x}$ and its realization is $x$.

## 2 A GRAPHICAL PROBABILITY FRAMEWORK FOR ATTENTION-MOE

Probabilistic graphical models (PGMs) provide a framework to understand conditional dependencies of and perform inference on variables. In this section, we demonstrate that multihead attention in MoE can be interpreted as a point estimate of a three-layer hierarchical mixture of expert regression. From this graphical model perspective, we uncover the underlying assumptions about the conditional independence between variables, highlighting the limitations that can arise from these assumptions.

### 2.1 CONNECTION BETWEEN MULTIHEAD ATTENTION AND 2-LAYER HIERARCHICAL MOE REGRESSIONS

This section shows that multihead-attention is the regression function of a two-layer hierarchical mixture of expert regressions. Considering a regression problems, which given any sequence $\tilde{\mathbf{X}} = [\tilde{\mathbf{x}}_1, \ldots, \tilde{\mathbf{x}}_N]^T$, we want to form a prediction $\boldsymbol{f}(\tilde{\mathbf{X}})$ of the target sequence variable $\tilde{\mathbf{O}} =$

$[\tilde{\mathbf{o}}_1, \ldots, \tilde{\mathbf{o}}_N]^T$. Suppose that by doing so, we incur the average square loss:

$$\inf_{\boldsymbol{f}} \mathbb{E}[L(\tilde{\mathbf{X}}, \tilde{\mathbf{O}})] = \inf_{\boldsymbol{f}} \int \|\boldsymbol{f}(\mathbf{X}) - \mathbf{O}\|_F^2 \, p(\mathbf{X}, \mathbf{O}) \, d\boldsymbol{x}_1 \ldots d\boldsymbol{x}_N \, d\boldsymbol{o}_1 \ldots d\boldsymbol{o}_N,$$

where $p(\mathbf{X}, \mathbf{O})$ is a joint density of the distribution of $\mathbf{X}$ and $\mathbf{O}$. The optimal regression function is

$$\boldsymbol{f}^\star(\mathbf{X}) = \mathbb{E}[\tilde{\mathbf{O}} \mid \tilde{\mathbf{X}} = \mathbf{X}] = [\mathbb{E}[\tilde{\mathbf{o}}_1 \mid \tilde{\mathbf{X}} = \mathbf{X}], \ldots, \mathbb{E}[\tilde{\mathbf{o}}_N \mid \tilde{\mathbf{X}} = \mathbf{X}]]. \tag{3}$$

As a result, we are interested in obtaining $\boldsymbol{f}_i^\star(\mathbf{X}) = \mathbb{E}[\tilde{\mathbf{o}}_i \mid \mathbf{X}]$ for $i = 1, \ldots, N$. The detailed derivation of the optimal regression function is given in Appendix B.

**Multihead Attention as a special case**. We present a graphical model $\mathcal{G}_1$ in Figure 1 (Left). Let us define the following variables: $\tilde{\mathbf{u}}_i$ is the output variable of interest, $\tilde{z}_i = (h', j)$ is the index of the attention head and position, $\tilde{h}_i = h$ is the index of the selected head for each $i$, and $\tilde{\mathbf{x}}_j$ is the input variable observed at position $j$. The graphical model $\mathcal{G}_1$ admits the following generating process: 1. $\mathbb{P}(\tilde{h}_i = h \mid \tilde{\mathbf{x}}_i) = \frac{1}{H}$, for all $h = 1, \ldots, H$, 2. $\mathbb{P}(\tilde{z}_i = (h', j) \mid \tilde{h}_i = h, \mathbf{X}) = \mathrm{softmax}\left((\mathbf{W}_{Q,h} x_i)^\top \mathbf{W}_{K,h} \boldsymbol{x}_j / \sqrt{D_{qk}}\right) \mathbb{I}(h' = h)$, 3. $\mathbb{P}(\tilde{\mathbf{u}}_i \mid \tilde{z}_i = (h', j), \tilde{\mathbf{x}}_j = \boldsymbol{x}_j) = \mathcal{N}(\tilde{\mathbf{u}}_i | \mathbf{W}_{O,h'} \mathbf{W}_{V,h'} \boldsymbol{x}_j, \sigma^2 \mathbf{I})$, where $\sigma > 0$ is a standard deviation scalar.

Now, we use $\tilde{\mathbf{U}}$ as our target variable i.e., $\tilde{\mathbf{U}}$ plays the role of $\tilde{\mathbf{O}}$ in Equation 3. As discussed, our goal is to estimate $\mathbb{E}[\tilde{\mathbf{u}}_i \mid \tilde{\mathbf{X}} = \mathbf{X}]$ for $i = 1, \ldots, N$, the conditional expectation of $\tilde{\mathbf{u}}_i$ given $\tilde{\mathbf{X}} = \mathbf{X}$. Using the tower rule, we take expectations over $\tilde{\mathbf{u}}_i$ conditioned on $\tilde{z}_i$, $\tilde{h}_i$, and $\tilde{\mathbf{X}}$:

$$\mathbb{E}[\tilde{\mathbf{u}}_i \mid \tilde{\mathbf{X}}] = \mathbb{E}\big[\mathbb{E}\big[\mathbb{E}[\tilde{\mathbf{u}}_i \mid \tilde{z}_i, \tilde{h}_i, \tilde{\mathbf{X}}] \mid \tilde{h}_i, \tilde{\mathbf{X}}\big] | \tilde{\mathbf{X}}\big] = \frac{1}{H} \sum_{h=1}^{H} \mathbf{W}_{O,h} \sum_{j=1}^{N} \mathrm{softmax}\left(\frac{\mathbf{q}_{i,h}^\top \mathbf{k}_{j,h}}{\sqrt{D}}\right) \mathbf{v}_{j,h},$$

which is the multihead attention (see Equation 1). The detailed derivation is given in Appendix B.

## 2.2 Attention-(S)MoE as a Point Estimate of a three-layer Hierarchical MoE

In this section, we show that the attention-MoE is a point estimate of the regression function in a three-layer hierarchical mixture of experts, and attention-SMoE is its sparse version. We discuss further the graphical model $\mathcal{G}_1$. Let $\tilde{e}_i \in \{1, \ldots, K\}$ represent the expert assigned to token $i$. Each token variable $\tilde{\mathbf{u}}_i$ can choose one of $K$ experts.

**Probabilistic graphical model.** The graphical model $\mathcal{G}_1$ admits the following generating process after the generation in the previous session (Section 2.1): 4. The probability of selecting expert $k$ for token $i$, given its embedding $\boldsymbol{u}_i$, is determined by a softmax function: $\mathbb{P}(\tilde{e}_i = k \mid \tilde{\mathbf{u}}_i = \boldsymbol{u}_i) = \mathrm{softmax}(\boldsymbol{u}_i^\top \mathbf{W}[k] + \boldsymbol{b}[k])$, where $\mathbf{W}, \boldsymbol{b}$ are defined in Section 1.2, 5. The output $\tilde{\mathbf{o}}_i$ conditioned on the expert assignment $\tilde{e}_i = k$ and the token embedding $\tilde{\mathbf{u}}_i = \boldsymbol{u}_i$ follows a Gaussian distribution: $\mathbb{P}(\tilde{\mathbf{o}}_i \mid \tilde{\mathbf{u}}_i = \boldsymbol{u}_i, \tilde{e}_i = k) \sim \mathcal{N}(\tilde{\mathbf{o}}_i \mid \mathbf{g}_k(\boldsymbol{u}_i), \mathbf{I})$, where $\mathbf{g}_k(\boldsymbol{u}_i)$ is the expert-specific function.

**Optimal regression function.** Now, our goal is to estimate the conditional expectation of the output $\tilde{\mathbf{o}}_i$ given token $\tilde{\mathbf{X}}$ i.e., $\mathbb{E}[\tilde{\mathbf{o}}_i \mid \mathbf{X}]$. Using the tower rule, we compute the expectation over $\tilde{\mathbf{o}}_i$ by conditioning on $\tilde{\mathbf{u}}_i$, $\tilde{e}_i$, and $\tilde{\mathbf{X}}$. Following the PGM $\mathcal{G}_1$ we have $(\tilde{\mathbf{o}}_i \perp\!\!\!\perp \tilde{\mathbf{X}} \mid \tilde{\mathbf{u}}_i, \tilde{e}_i)$, and $(\tilde{e}_i \perp\!\!\!\perp \tilde{\mathbf{X}} \mid \tilde{\mathbf{u}}_i)$. We obtain:

$$\mathbb{E}[\tilde{\mathbf{o}}_i \mid \mathbf{X}] = \mathbb{E}\big[\mathbb{E}[\mathbb{E}[\tilde{\mathbf{o}}_i \mid \tilde{e}_i, \tilde{\mathbf{u}}_i] \mid \tilde{\mathbf{u}}_i] | \tilde{\mathbf{X}}\big] = \mathbb{E}\left[\sum_{k=1}^{K} \mathrm{softmax}(\boldsymbol{u}_i^\top \mathbf{W}_k) \mathbf{g}_k(\boldsymbol{u}_i) | \tilde{\mathbf{X}}\right]. \tag{4}$$

We obtain final equality in (4) since $\mathbb{E}[\tilde{\mathbf{o}}_i \mid \tilde{\mathbf{u}}_i = \boldsymbol{u}_i, \tilde{e}_i = k] = \mathbf{g}_k(\boldsymbol{u}_i)$. Attention-MoE approximates this expectation by using a constant estimate of $\tilde{\mathbf{u}}_i$ given $\mathbf{X}$, rather than fully marginalizing over the distribution of $\boldsymbol{u}_i$. This simplification results in a discriminant block and introduces a biased estimate:

$$\bar{\mathbf{o}}_i = \mathbb{E}[\tilde{\mathbf{o}}_i \mid \tilde{\mathbf{u}}_i = \mathbb{E}[\tilde{\mathbf{u}}_i \mid \mathbf{X}]] = \mathrm{MoE}(\mathrm{MHA}(\mathbf{X})[i]),$$

where the MoE block leverages the output of the multi-head attention (MHA) to approximate the latent representation $\tilde{\mathbf{u}}_i$.

**Remark.** From this interpretation, Attention-SMoE is a special case of the general MoE where $\mathbb{P}(\tilde{e}_i = k \mid \tilde{\mathbf{u}}_i = \boldsymbol{u}_i) = \mathrm{TopM\_Renormalize}(\mathrm{softmax}(\mathbf{W}\tilde{\mathbf{u}}_i + \boldsymbol{b}))[k]$. Therefore, we can formulate

the probabilistic graphical model (PGM) of MoE in general terms, with SMoE following as a specific instance of this formulation.

From the graphical model $\mathcal{G}_1$, we observe that expert selections for each individual token are conditionally independent given the tokens, meaning $(\tilde{e}_i \perp\!\!\!\perp \tilde{e}_j | \tilde{\mathbf{X}})$ for all $i, j$. This lack of interaction between tokens' decisions can lead to *routing fluctuation*. To elaborate, at the end of training, when the learning rate is significantly small and the model parameters stabilize, we expect minimal changes in routing decisions, given that the approximation function is reasonably smooth and token representations do not change considerably. However, empirical evidence shows this is not the case. In Section 4, we present an empirical analysis demonstrating that upto **33% of tokens still switch their assigned experts in the final epochs**, highlighting a persistent instability in routing. This observation suggests that similar tokens should be routed to the same expert, but the current independent routing does not guarantee this from happening. By letting tokens influencing others' expert selection, we could reduce fluctuation and ensure more stable, consistent routing decisions.

# 3 MUTUAL-INFORM (S)MoEs

Leveraging token similarity to guide expert selection can both reduce routing fluctuations and facilitate expert learning by presenting less diverse input to each expert. In this section, from the PGM perspective, we introduce the notion of Mutual-Inform MoEs, where expert decisions are directly dependent based on tokens' correspondence. These correspondence can be computed directly from token embeddings $\tilde{\mathbf{u}}_i$ within the MoE layers, given the Similarity-Inform mechanism (Section 3.1) or derived from the attention layer, a variation we call Attention-Informed MoE (Section 3.2). Their sparse version, Similarity-Infom SMoEs and Attention-Inform SMoE are derived accordingly as special cases, forming the notion of Mutual-Inform SMoEs. In Section 3.3, we present an entropy analysis of the Mutual-Inform MoE model to highlight the advantage of our method in reducing routing fluctuation. Specifically, we demonstrate how our approach lowers the entropy of indecisive or less confident tokens, making them less prone to fluctuations in their routing decisions.

## 3.1 TOKEN ROUTING WITH SIMILARITY-INFORM

This section introduces a novel approach to token routing that leverages token similarities to inform decision-making, addressing the limitations of independent routing observed in SMoEs.

**Probabilistic graphical model.** We present a PGM $\mathcal{G}_2$ in Figure 1 (Middle) that encapsulates the conditional dependencies of expert decisions for tokens. We introduce a similarity variable $\tilde{s}_i$ that quantifies the likelihood of token $\tilde{\mathbf{u}}_i$ being similar to other tokens $\tilde{\mathbf{u}}_j$. After a similar generative process in Section 2.1 and step 4 in Section 2.2 , $\mathcal{G}_2$ admit the following additional generation:

5. The similarity is computed using a scaled dot-product attention mechanism:

$$\mathbb{P}(\tilde{s}_i = j \mid \tilde{\mathbf{U}} = \mathbf{U}) = \text{softmax}\left(\frac{\boldsymbol{u}_i^T \mathbf{W}_s \boldsymbol{u}_j}{\tau}\right) \tag{5}$$

where $\mathbf{W}_s$ is a learnable parameter matrix and $\tau > 0$ is a temperature parameter controlling the sharpness of the similarity distribution.

6. The final expert decision for token $i$ is then defined as:

$$\mathbb{P}(\tilde{d}_i = k \mid \tilde{\mathbf{e}} = \mathbf{e}, \tilde{s}_i = j) = \mathbb{I}(k = e_j). \tag{6}$$

Here, $\tilde{\mathbf{e}} = [\tilde{e}_1, \ldots, \tilde{e}_n]^T$ represents the choice of expert for all token for token $i$ with $e_i$ as its realization. Equation 6 implies that similar tokens are more likely to be routed to the same expert, promoting consistency in the processing of related information.

7. As in Section 2.2, we assume that $\mathbb{P}(\tilde{\mathbf{o}}_i \mid \tilde{\mathbf{u}}_i = \boldsymbol{u}_i, \tilde{d}_i = k) = \mathcal{N}(\tilde{\mathbf{o}}_i \mid \mathbf{g}_k(\boldsymbol{u}_i), \mathbf{I})$.

**Optimal regression function.** To determine the best prediction for each token $\tilde{\mathbf{o}}_i$, given $\tilde{\mathbf{X}}$, we compute the expectation $\mathbb{E}[\tilde{\mathbf{o}}_i \mid \tilde{\mathbf{X}}]$. Unlike previous cases, here we condition on $\tilde{\mathbf{U}}$, $\tilde{d}_i$ and $\tilde{\mathbf{X}}$, as the decision $\tilde{d}_i$ is not independent of $\tilde{\mathbf{u}}_j$ given $\tilde{\mathbf{u}}_i$. In addtion, from the PGM $\mathcal{G}_2$, we have

$\tilde{\mathbf{o}}_i \perp\!\!\!\perp \{\tilde{\mathbf{u}}_j\}_{j \neq i} \mid (\tilde{d}_i, \tilde{\mathbf{u}}_i)$. Thus:

$$\mathbb{E}[\tilde{\mathbf{o}}_i \mid \tilde{\mathbf{X}}] = \mathbb{E}\Big[\mathbb{E}\big[\mathbb{E}[\tilde{\mathbf{o}}_i \mid \tilde{d}_i, \tilde{\mathbf{u}}_i = \boldsymbol{u}_i] \mid \tilde{\mathbf{U}}\big] \mid \tilde{\mathbf{X}}\Big] = \mathbb{E}\left[\sum_{k=1}^{K} \mathbb{P}(\tilde{d}_i = k \mid \tilde{\mathbf{U}})\mathbf{g}_k(\boldsymbol{u}_i) \mid \tilde{\mathbf{X}}\right]. \qquad (7)$$

From Equation (4), the probability of final expert assignment given $\tilde{\mathbf{U}}$ of token $i$ is

$$\mathbb{P}(\tilde{d}_i = k \mid \tilde{\mathbf{U}}) = \sum_{j=1}^{N} \sum_{e_1=1}^{K} \cdots \sum_{e_N=1}^{K} \mathbb{P}(\tilde{d}_i = k \mid \tilde{\mathbf{e}} = \mathbf{e}, \tilde{s}_i = j) \prod_{i'=1}^{N} \mathbb{P}(\tilde{e}_{i'} = e_{i'} \mid \tilde{\mathbf{u}}_i)\mathbb{P}(\tilde{s}_i = j \mid \tilde{\mathbf{U}})$$

$$= \sum_{j=1}^{N} \mathbb{P}(\tilde{e}_j = k \mid \tilde{\mathbf{u}}_j)\mathbb{P}(\tilde{s}_i = j \mid \tilde{\mathbf{U}}). \qquad (8)$$

Substitude into Equation (5), we get:

$$\mathbb{E}[\tilde{\mathbf{o}}_i \mid \tilde{\mathbf{X}}] = \mathbb{E}\left[\sum_{k=1}^{K} \sum_{j=1}^{N} \mathbb{P}(\tilde{e}_j = k \mid \tilde{\mathbf{u}}_j)\mathbb{P}(\tilde{s}_i = j \mid \tilde{\mathbf{U}})\mathbf{g}_k(\boldsymbol{u}_i) \mid \tilde{\mathbf{X}}\right]. \qquad (9)$$

Similar to MoE-transformer block, a point estimate of the regression function in Equation (7) can be obtained by conditioning on the point $\tilde{\mathbf{U}} = \mathbb{E}[\tilde{\mathbf{U}} \mid \tilde{\mathbf{X}} = \mathbf{X}] = \text{MHA}(\mathbf{X}) = \bar{\mathbf{U}} = [\bar{\mathbf{u}}_1, \ldots, \bar{\mathbf{u}}_N]^T$

$$\bar{\mathbf{o}}_i = \sum_{k=1}^{K} \sum_{j=1}^{N} \mathbb{P}(\tilde{e}_j = k \mid \bar{\mathbf{u}}_j])\mathbb{P}(\tilde{s}_i = j \mid \bar{\mathbf{U}})\mathbf{g}_k(\bar{\mathbf{u}}_i). \qquad (10)$$

**Similarity-Inform (S)MoE.** With the previous results, we now define Similarity-Inform (S)MoE:

**Definition 1.** *(Similarity-Inform SMoE) Given a input sequence of tokens input* $\mathbf{X}$*, the output of the multi-head attention layer is* $\bar{\mathbf{U}} = \text{MHA}(\mathbf{X}) = [\bar{\mathbf{u}}_1, \ldots, \bar{\mathbf{u}}_N]^T$*, the normalized expert score* $\mathbf{e}_i = [\text{softmax}(r_1(\bar{\mathbf{u}}_i)), \ldots, \text{softmax}(r_K(\bar{\mathbf{u}}_i))]$ *for each token* $i$ *and the similarity score* $\mathbf{S}[i, j] = \text{softmax}\left(\bar{\mathbf{u}}_i^T \mathbf{W}_s \bar{\mathbf{u}}_j / \tau\right)$*, Similarity-Inform MoE computes the output* $\bar{\mathbf{o}}_i$ *at token* $i$ *as*

$$\bar{\mathbf{o}}_i = \sum_{k=1}^{K} \sum_{j=1}^{N} \mathbf{S}[i, j]\mathbf{e}_j[k]\mathbf{g}_k(\bar{\mathbf{u}}_i).$$

*and its special version Similarity-Inform SMoE calculates*

$$\bar{\mathbf{o}}_i = \sum_{k=1}^{K} \text{TopM\_Renormalize}\big(\sum_{j=1}^{N} \mathbf{S}[i, j]\mathbf{e}_j\big)[k]\mathbf{g}_k(\bar{\mathbf{u}}_i). \qquad (11)$$

By incorporating token similarities, encourages experts to specialize in handling clusters of similar tokens, leading to more efficient learning and better performance. In addition, the approach is less likely to make drastically different routing decisions for similar tokens, hence, leads to reduction in routing fluctuations and improve robustness.

## 3.2 Token Routing with Attention-Inform

The routing decision for each token can also be informed via their dependency capture in the attention layers. Instead of directly basing the final decision $\tilde{d}_i$ of each token on the similarity variable $\tilde{s}_i$ as Similarity-Inform (S)MoEs, we establish a link from the variable $z_i$ — which represents the token that token $i$ attends to in the attention layer — to the decision $\tilde{d}_i$. In this way, rather than computing the similarity matrix based on $\bar{\mathbf{U}}$, the input of SMoE layers, we utilize the similarity information directly from the attention layer to inform the expert choice for tokens. This approach also leads to a consistent decision process because the attention layers inherently capture the relationships between tokens. This means the choice of which expert a token is routed to is aligned with the token's interactions in the attention mechanism.

**Probabilistic graphical model.** The method is presented in the PGM $\mathcal{G}_3$ in Figure 1 (Right), which shares a similar generative process (Step 1, 2, 3) in Section 2.1 and Step 4 in Section 2.2, with the following additional generation:

5. Unlike the graph $\mathcal{G}_2$, under the graph $\mathcal{G}_3$, $\tilde{d}_i$ is no longer conditional independent of $\tilde{\mathbf{X}}$ given $\tilde{\mathbf{U}}$ i.e., $\mathbb{P}(\tilde{d}_i = k \mid \tilde{\mathbf{e}} = \mathbf{e}, \tilde{z}_i = (h', j)) = \mathbb{I}(k = e_j)$.

**Optimal regression function.** The best guess of $\tilde{\mathbf{o}}_i$ given $\tilde{\mathbf{X}}$ in Equation 5 becomes:

$$\mathbb{E}[\tilde{\mathbf{o}}_i \mid \tilde{\mathbf{X}}] = \mathbb{E}\left[\sum_{k=1}^{K} \mathbb{P}(\tilde{d}_i = k \mid \tilde{\mathbf{U}}, \tilde{\mathbf{X}})\mathbf{g}_k(\boldsymbol{u}_i)] \mid \tilde{\mathbf{X}}\right] \tag{12}$$

Lemma 1 provides the key result for computing this expectation. The details derivation of Lemma 1 is found in Appendix A.1

**Lemma 1.** *The dependency of $\tilde{d}_i$ on $\tilde{\mathbf{U}}$ and $\tilde{\mathbf{X}} = [\boldsymbol{x}_1, \ldots, \boldsymbol{x}_N]^T$ is given by*

$$\mathbb{P}(\tilde{d}_i = k \mid \tilde{\mathbf{U}}, \tilde{\mathbf{X}}) = \sum_{h=1}^{H}\sum_{j=1}^{N} \mathbb{P}(\tilde{e}_j = k \mid \tilde{\mathbf{u}}_j)\mathbb{P}(\tilde{z}_i = (h, j) \mid \tilde{h}_i = h, \tilde{\mathbf{u}}_i, \tilde{\mathbf{X}})\mathbb{P}(\tilde{h}_i = h \mid \tilde{\mathbf{u}}_i, \tilde{\mathbf{X}})$$

$$= \sum_{h=1}^{H}\sum_{j=1}^{N} \mathbf{H}'[i, h]\mathbf{A}'_h[i, j]\mathbf{E}[j, k], \tag{13}$$

*where $\mathbf{E}[j, k] = \mathbb{P}(\tilde{e}_i = k \mid \tilde{\mathbf{u}}_i = \boldsymbol{u}_i)$ and the posteriors*

$$\mathbf{A}'_h[i, j] := \mathbb{P}(\tilde{z}_i = (h, j) \mid \tilde{h}_i = h, \tilde{\mathbf{u}}_i, \tilde{\mathbf{X}}) = \frac{\mathbf{A}_h[i, j]\mathcal{N}(\tilde{\mathbf{u}}_i \mid \mathbf{W}_{O,h}\mathbf{W}_{V,h}\boldsymbol{x}_j, \sigma^2\mathbf{I})}{\sum_{j'} \mathbf{A}_h[i, j']\mathcal{N}(\tilde{\mathbf{u}}_i \mid \mathbf{W}_{O,h}\mathbf{W}_{V,h}\boldsymbol{x}_{j'}, \sigma^2\mathbf{I})},$$

$$\mathbf{H}'[i, h] := \mathbb{P}(\tilde{h}_i = h \mid \tilde{\mathbf{u}}_i, \mathbf{X}) = \frac{\mathbf{H}[i, h]\sum_j \mathbf{A}_h[i, j]\mathcal{N}(\tilde{\mathbf{u}}_i \mid \mathbf{W}_{O,h}\mathbf{W}_{V,h}\boldsymbol{x}_j, \sigma^2\mathbf{I})}{\sum h'\mathbf{H}[i, h']\sum_{j'} \mathbf{A}_{h'}[i, j']\mathcal{N}(\tilde{\mathbf{u}}_i \mid \mathbf{W}_{O,h'}\mathbf{W}_{V,h'}\boldsymbol{x}_{j'}, \sigma^2\mathbf{I})},$$

*with the prior $\mathbf{A}_h[i, j] = \mathbb{P}(\tilde{z}_i = (h, j) \mid \tilde{h}_i = h, \mathbf{X})$ and $\mathbf{H}[i, h] = \mathbb{P}(\tilde{h}_i = h \mid \boldsymbol{x}_i)$.*

Lemma 1 unveils a sophisticated decision-making process in the Attention-Inform MoE, where the final routing decision for a token is influenced by the decisions of other tokens as well as the relevance of each attention head. This formulation can be interpreted as a two-stage influence process: First, each token's original decision is adjusted by the decisions of other tokens, weighted by $A'_h[i, j]$, which represents the "responsibility" of token $j$ in explaining token $i$'s representation within attention head $h$. Then, these weighted decisions from each head are further weightedly combined by $H'[i, h]$, which represents the responsibility of head $h$ in explaining token $i$. This hierarchical weighting scheme allows the model to integrate context from multiple attention patterns.

Substitute the results of Lemma 1 to Equation (12), given $\tilde{\mathbf{X}} = \mathbf{X}$, the best guess of $\tilde{\mathbf{o}}_i$ is obtained as

$$\mathbb{E}[\tilde{\mathbf{o}}_i \mid \tilde{\mathbf{X}}] = \mathbb{E}\left[\sum_{h=1}^{H}\sum_{k=1}^{K}\sum_{j=1}^{N} \mathbf{H}'[i, h]\mathbf{A}'_h[i, j]\mathbf{E}[j, k]\mathbf{g}_k(\boldsymbol{u}_i) \mid \tilde{\mathbf{X}}\right]. \tag{14}$$

To mitigate the computational cost of full posterior inference across all heads, we propose an approximation that enhances posterior certainty while reducing computational overhead. For all $i = 1, \ldots, N$, we approximate $\mathbf{H}'[i, h]$ as follows:

$$\bar{\mathbf{H}}[i, h] = \mathbb{I}(h = h^* := \arg\min_h \mathbb{E}[\mathcal{H}(\mathbf{A}_h[i])]), \tag{15}$$

where $\mathcal{H}(\mathbf{A}_h[i])$ is the entropy of attention score for token $i$ at head $h$ and the expectation $\mathbb{E}[\mathcal{H}(\mathbf{A}_h[i])]$ is taken over tokens $i$. This means that only the attention head with the lowest average entropy should contribute to the posteriors. The final point estimate of the regression function in Equation (12) can be obtained by conditioning on the point $\tilde{\mathbf{U}} = \mathbb{E}[\tilde{\mathbf{U}} \mid \tilde{\mathbf{X}} = \mathbf{X}]$, resulting in

$$\bar{\mathbf{o}}_i = \sum_{k=1}^{K}\sum_{j=1}^{N} \mathbf{A}'_{h^*}[i, j]\bar{\mathbf{e}}_j[k]\mathbf{g}_k(\bar{\mathbf{u}}_i), \tag{16}$$

where $\bar{\mathbf{e}}_j[k] = \mathbb{P}(\tilde{e}_i = k \mid \tilde{\mathbf{u}}_i = \bar{\mathbf{u}}_i)$.

**Attention-Inform (S)MoE.** With the previous results, we define Attention-Inform (S)MoE.

**Definition 2.** *(Attention-Inform (S)MoE) Given an input sequence of tokens $\mathbf{X}$, the output of the multihead attention layer is $\bar{\mathbf{U}} = \mathrm{MHA}(\mathbf{X}) = [\boldsymbol{u}_1, \dots, \boldsymbol{u}_N]^T$, the normalized expert score $\bar{\mathbf{e}}_i = [\mathrm{softmax}(r_1(\bar{\mathbf{u}}_i)), \dots, \mathrm{softmax}(r_K(\bar{\mathbf{u}}_i))]$ for each token $i$ and the posterior score $\mathbf{A}'_{h^*}$ computed as in Definition 1 with $h^*$ being the head index with lowest average attention entropy defined in Equation (19), the Attention-Inform MoE computes the output $\bar{\mathbf{o}}_i$ at token $i$ as in Equation (16) while its SMoE version, the Attention-Inform SMoE computes*

$$\bar{\mathbf{o}}_i = \sum_{k=1}^{K} \mathrm{TopM\_Renormalize}\big(\sum_{j=1}^{N} \mathbf{A}'_{h^*}[i,j]\bar{\mathbf{e}}_j\big)[k]\mathbf{g}_k(\bar{\mathbf{u}}_i). \tag{17}$$

### 3.3 ON THE ENTROPY ANALYSIS OF MUTUAL-INFORM MOE

When the model is uncertain in its routing decision, a small perturbation in either weight space or input space would cause a change in its discrete decision. As a result, high entropy in expert selection scores of a token suggests increased routing fluctuation in SMoE. In this section, we demonstrate that Mutual-Inform MoE reduces routing fluctuations by lowering the entropy of routing scores.

For any $i = 1, \dots, N$, and define $J_i = \{j \mid \mathcal{H}(\tilde{e}_j \mid \tilde{\mathbf{u}}_j) \leq \mathcal{H}(\tilde{e}_i \mid \tilde{\mathbf{u}}_i)\}$. Here, we slightly abuse the notation of entropy $\mathcal{H}$, using it interchangeably for both a random variable and its associated distribution.. Let Mutual-Inform MoE be applied to token $i$ with the set $J_i$. The score function $s(i,j)$, capturing the correspondence between token $i$ and $j \in J_i$, is either defined as $s(i,j) = \mathrm{softmax}\big(\bar{\mathbf{u}}_i^T \mathbf{W}_s \bar{\mathbf{u}}_j / \tau\big)$, or $s(i,j) = \mathbf{A}'_{h^*}[i,j]$ from Lemma 1. We show that the Mutual Inform MoE provides an upper bound in the entropy of the weighted decision:

**Proposition 1.** *Let $\mathbf{p}_i = [p_1, \dots, p_K]^T$ be the distribution of the final decision variable $\tilde{d}_i$, representing the final routing score of token $i$. Whereas the original routing score of token $i$, as defined in Definition 1, is denoted as $\bar{\mathbf{e}}_i$. Applying Mutual-Inform MoE to recalculate the tokens' decision score yields $\mathbf{p}_i = \sum_{j=1}^{|J_i|} s(i,j)\bar{\mathbf{e}}_j$. Thus, the upper bound of entropy of the final decision is given by:*

$$\mathcal{H}(\mathbf{p}_i) \leq \sum_{j=1}^{|J_i|} s(i,j)\mathcal{H}(\bar{\mathbf{e}}_j) + \mathcal{H}(\mathbf{s}_i), \tag{18}$$

*where $\mathbf{s}_i = [s(i,1), \dots, s(i,|J_i|)]^T$. And as $\tau \to 0$ (for Similarity-Inform) or $\sigma \to 0$ (for Attention-Inform), $\mathcal{H}(\mathbf{p}_i) \leq \mathcal{H}(\bar{\mathbf{e}}_i)$.*

In Proposition 1, $\sigma$ is the standard deviation of $\bar{\mathbf{u}}_i$, which affects $s(i,j) = \mathbf{A}'_{h^*}[i,j]$ as defined in Lemma 1. The proof of Proposition 1 is given in Appendix A.2. Mutual-Inform MoE approach can effectively reduce routing fluctuation in Sparse Mixture of Experts (SMoE) models by lowering the entropy of the routing scores. A high entropy in the expert choices indicates uncertainty in token routing, which can lead to fluctuations in the routing decisions. Our approach provides an upper bound on the entropy of the final decision for each token, showing that as the temperature approaches zero, the entropy of the final decision reduces compared to the entropy of the original routing score. Thus, the model can improve its decision certainty, reducing the fluctuation in token routing. In practice, we relax constraints by letting $J_i = \{1, \dots, N\}$, where $N$ is the number of tokens.

## 4 EXPERIMENTAL RESULTS

To demonstrate the advantages of Mutual-Inform SMoE, we perform extensive experiments on ImageNet classification and Wikitext-103 language modeling. We further show the significant improvement in model robustness by evaluating the model with adversarially and naturally perturbed version of these datasets.

**Language modeling on Wikitext-103.** Table 1 highlights the significant performance and robustness improvements of our Mutual-Inform SMoEs compared to SMoE and GLAM (Generalist Language Model) (Du et al., 2022) baselines on the Wikitext-103 language modeling benchmark, using TopM

Table 1: PPL evaluation (lower is better) with the clean and attacked Wikitext-103 test set Baseline SMoE, Attention-Inform SMoE, and Similarity-Inform SMoE.

| Model/Metric | Clean Wikitext-103 | | Attacked Wikitext-103 | |
|---|---|---|---|---|
| | Valid PPL | Test PPL | Valid PPL | Test PPL |
| *SMoE (M = 1)* | 39.55 | 40.75 | 48.82 | 50.21 |
| Similarity-inform SMoE (M = 1) | 37.78 | 39.18 | 46.93 | 48.66 |
| Attention-inform SMoE (M = 1) | 38.02 | 39.35 | 47.20 | 48.72 |
| *SMoE (M = 2)* | 33.29 | 34.84 | 41.75 | 43.59 |
| Similarity-inform SMoE (M = 2) | **30.75** | **32.03** | **38.33** | **39.92** |
| Attention-inform SMoE (M = 2) | 31.31 | 32.23 | 39.68 | 40.91 |
| *GLAM (k = 2)* | 37.55 | 39.10 | 48.01 | 49.75 |
| Similarity-inform GLAM (M = 2) | 33.72 | 34.92 | 42.19 | 43.72 |
| Attention-inform GLAM (M = 2) | 35.17 | 36.71 | 44.17 | 45.85 |

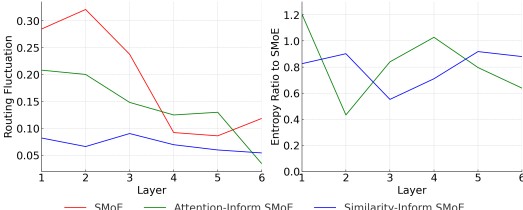

Figure 2: Comparison of routing fluctuation and entropy ratio across layers for Baseline SMoE, Attention-Inform SMoE, and Similarity-Inform SMoE

experts (M=1 or M=2). Performance is evaluated using perplexity scores on both the validation and test sets, where lower values indicate better model performance. Additionally, the table presents results from an adversarial scenario, where the dataset undergoes word swap attacks, allowing for a robust assessment of the models. Across all configurations, the proposed Mutual-Inform mechanisms, both Similarity and Attention-Inform SMoE, consistently outperform their baseline counterparts. Notably, the Similarity-Inform SMoE with $M = 2$ experts achieves the lowest perplexity scores on both clean and adversarial datasets, demonstrating its effectiveness in improving language modeling performance and resilience against adversarial perturbations. Similarly, GLAM-based models follow these trends, with Mutual-Inform variants showing considerable improvements over their standard implementations. These results underscore the clear advantage of the Mutual-Inform approach in both performance and robustness.

**ImageNet Classification.** Table 2 demonstrates the improvement in performance and robustness of our methods compared to the baseline V-MoE (Riquelme et al., 2021) model. Both the Similarity-Inform and Attention-Inform variants show consistent gains in the clean data and across consistently more robust than the DeiT baseline under other adversarial examples and out-of-distribution dataset, including the ImageNet-C (common data corruption and perturbations, such as adding noise and blurring the images) (Hendrycks & Dietterich, 2019), ImageNet-A (adversarial examples) (Hendrycks et al., 2021b), ImageNet-R (out of distribution generalization) (Hendrycks et al., 2021a), and ImageNet-O (out-of-distribution detection) (Hendrycks et al., 2021b) datasets.

Table 2: Test set accuracy of different ImageNet variants on Baseline SMoE, Attention-Inform SMoE, and Similarity-Inform SMoE. All SMoE models are trained only on the original ImageNet dataset.

| Model | Params | IN-1K | IN-R | IN-A | IN-C |
|---|---|---|---|---|---|
| | | Top-1 ↑ | Top-1 ↑ | Top-1 ↑ | Top-1 ↑ |
| *V-MoE (baseline)* | 297M | 72.71 | 35.42 | 5.27 | 48.72 |
| Similarity-Inform V-MoE | 297M | 73.21 | 36.58 | 5.60 | 50.45 |
| Attention-Inform V-MoE | 297M | **73.33** | **36.66** | **6.78** | **50.85** |

Next, we demonstrate the reduction in entropy and routing fluctuation of Mutual-Inform (S)MoEs, empirically on Wikitext-103 with top $M = 2$ experts. In addition, we conducted further analysis on the case of $M = 1$ in Appendix C.2, and visualize the load-balancing property in Appendix D.2.

**Mutual-Inform MoE reduces routing fluctuation** Figure 2 (Left) compares the routing fluctuation of the Baseline SMoE, Attention-Inform SMoE, and Similarity-Inform SMoE. The fluctuation rate, computed as the proportion of tokens that switch one or both expert choices between consecutive last training epochs (from epoch 59 to 60), provides insight into routing stability. The baseline SMoE exhibits the highest overall fluctuation rates, particularly in the initial layers. In contrast, both the

Attention-Inform and Similarity-Inform SMoE methods demonstrate markedly lower fluctuation rates across all layers. The Similarity-Inform SMoE, in particular, maintains consistently low fluctuation rates throughout the network, indicating better stability in routing decisions. The Attention-Inform SMoE shows an overall significant improvement in routing fluctuation over the baseline.

**Mutual-Inform MoE reduces decision entropy.** Figure 2 (Right) illustrates our models' average rate of entropy of tokens' routing decisions across layers to the baseline SMoE. This rate of entropy is computed for epoch 59, which is the epoch immediately preceding the final epoch where routing fluctuation is observed. Our proposed methods, Attention-Inform and Similarity-Inform SMoE, demonstrate lower average entropy compared to the baseline SMoE (the rate is smaller than 1). This trend aligns with the lower routing fluctuation observed in the left graph, suggesting that our approaches lead to more stable and consistent routing decisions. The Similarity-Inform SMoE, in particular, maintains lower entropy in the all layers, corresponding to its better stability in routing decisions. These results further demonstrates the advantage of our methods, leading to more consistent routing decision and model robustness.

## 5 RELATED WORK

Routing fluctuation as been discussed in existing literature. Nguyen et al. (2024) mentions that various SMoE routers Csordás et al. (2023); Do et al. (2023) suffer from routing fluctuation without proposing solutions. In addition Su et al. (2024) suggests that due to the variation of learnable parameters in the router. StableMoE (Dai et al., 2022) has been proposed to reduce the routing fluctuation problem by using two training stages. During the initial training phase, StableMoE develops a balanced and cohesive routing strategy, which it then distills into a lightweight router that operates independently of the backbone model. In the second training phase, StableMoE uses the distilled router to establish the token-to-expert assignments and locks this assignment in place to ensure a stable routing strategy. SMoE-dropout Chen et al. (2023) is another work that also provides another solution to improve the stability of the model. This method initially randomizes and freezes the router during training to provide stable routing strategies Zoph et al. (2022) examine several approaches to improve stability including removing multiplicative interactions, injecting model noise, and constraining activations and gradients. After the examination, the authors propose the router z-loss which enhance the training stability with no quality degradation. Chi et al. (2022) proposes to estimate the routing scores between tokens and experts on a low-dimensional hypersphere to achieve more consistent routing compared to the conventional approach. Feedforward layers are replaced by hash layers in (Roller et al., 2021) to to keep routing choices consistent. Lewis et al. (2021) formulates routing as a linear assignment problem that globally maximizes token-expert similarities for increasing the stability. Our work is orthogonal to these approaches: to reduce routing fluctuation, we encourage tokens to influence each other's routing decision based on their similarity.

## 6 CONCLUDING REMARKS

We have presented a probabilistic graphical model view point of attention-(S)MoE. From the new perspective, we have introduced a novel notion of (S)MoEs, named Mutual-Inform (S)MoEs, where expert decisions are made from all input tokens via their similarities and relevance. We have proposed two variants of Mutual-Inform (S)MoEs i.e., Similarity-Inform (S)MoE and Attention-Inform (S)MoE, in which a token's routing decision can be influenced by others'. We proved that our methods help improve confidence in the decision-making processes by reducing the entropy of expert assignments. Finally, we carry out extensive experiments on ImageNet classification and Wikitext-103 language modeling to demonstrate the advantages and robustness of our Mutual-Inform (S)MoE models. A limitation of our paper is that we have not considered a generative model that capture the token generation process in our PGM. Studying transformer-MoE from a generative model perspective is an exciting research direction. We leave it for future work.

**Reproducibility Statement:** Source codes for our experiments are provided in the supplementary materials of the paper. The details of our experimental settings and computational infrastructure are given in Section 4 and the Appendix. All datasets that we used in the paper are published, and they are easy to find in the Internet.

**Ethics Statement:** Given the nature of the work, we do not foresee any negative societal and ethical impacts of our work.

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

# Supplement to "Mutual-Inform SMoE: Improving Routing Stability via Probabilistic Graphical Model Interpretation"

**Table of Contents**

## A  TECHINCAL PROOFS

**Notation**: Given an event $A$, we use $\mathbb{P}(A)$ to denote the probability of event $A$. We use $\mathbb{E}[\tilde{\mathbf{X}}]$ to denote the expectation of $\tilde{\mathbf{X}}$

### A.1  PROOF OF LEMMA 1

**Restate Lemma 1**

**Lemma 1.** *The dependency of $\tilde{d}_i$ on $\tilde{\mathbf{U}}$ and $\tilde{\mathbf{X}} = [\boldsymbol{x}_1, \dots, \boldsymbol{x}_N]^T$ is given by*

$$\mathbb{P}(\tilde{d}_i = k \mid \tilde{\mathbf{U}}, \tilde{\mathbf{X}}) = \sum_{h=1}^{H} \sum_{j=1}^{N} \mathbb{P}(\tilde{e}_j = k \mid \tilde{\mathbf{u}}_j) \mathbb{P}(\tilde{z}_i = (h, j) \mid \tilde{h}_i = h, \tilde{\mathbf{u}}_i, \tilde{\mathbf{X}}) \mathbb{P}(\tilde{h}_i = h \mid \tilde{\mathbf{u}}_i, \tilde{\mathbf{X}})$$

$$= \sum_{h=1}^{H} \sum_{j=1}^{N} \mathbf{H}'[i, h] \mathbf{A}'_h[i, j] \mathbf{E}[j, k],$$

*where $\mathbf{E}[j, k] = \mathbb{P}(\tilde{e}_i = k \mid \tilde{\mathbf{u}}_i = \boldsymbol{u}_i)$ and the posteriors*

$$\mathbf{A}'_h[i, j] := \mathbb{P}(\tilde{z}_i = (h, j) \mid \tilde{h}_i = h, \tilde{\mathbf{u}}_i, \tilde{\mathbf{X}}) = \frac{\mathbf{A}_h[i, j] \mathcal{N}(\tilde{\mathbf{u}}_i \mid \mathbf{W}_{O,h} \mathbf{W}_{V,h} \boldsymbol{x}_j, \sigma^2 \mathbf{I})}{\sum_{j'} \mathbf{A}_h[i, j'] \mathcal{N}(\tilde{\mathbf{u}}_i \mid \mathbf{W}_{O,h} \mathbf{W}_{V,h} \boldsymbol{x}_{j'}, \sigma^2 \mathbf{I})},$$

$$\mathbf{H}^{'}[i,h] := \mathbb{P}(\tilde{h}_i = h \mid \tilde{\mathbf{u}}_i, \tilde{\mathbf{X}}) = \frac{\mathbf{H}[i,h] \sum_j \mathbf{A}_h[i,j] \mathcal{N}(\tilde{\mathbf{u}}_i \mid \mathbf{W}_{O,h}\mathbf{W}_{V,h}\boldsymbol{x}_j, \sigma^2\mathbf{I})}{\sum h' \mathbf{H}[i,h'] \sum_{j'} \mathbf{A}_{h'}[i,j'] \mathcal{N}(\tilde{\mathbf{u}}_i \mid \mathbf{W}_{O,h'}\mathbf{W}_{V,h'}\boldsymbol{x}_{j'}, \sigma^2\mathbf{I})},$$

*with the prior* $\mathbf{A}_h[i,j] = \mathbb{P}(\tilde{z}_i = (h,j) \mid \tilde{h}_i = h, \tilde{\mathbf{X}})$ *and* $\mathbf{H}[i,h] = \mathbb{P}(\tilde{h}_i = h \mid \boldsymbol{x}_i)$.

**Review the generalization of the graph** $\mathcal{G}_3$. The method is presented in the PGM $\mathcal{G}_3$ in Figure 1 (Right), with the following generative process:

1. $\mathbb{P}(\tilde{h}_i = h \mid \tilde{\mathbf{x}}_i) = \frac{1}{H}$, for all $h = 1, \ldots, H$.

2. $\mathbb{P}(\tilde{z}_i = (h', j) \mid \tilde{h}_i = h, \mathbf{X}) = \text{softmax}\left((\mathbf{W}_{Q,h}x_i)^\top \mathbf{W}_{K,h}\boldsymbol{x}_j / \sqrt{D_{qk}}\right) \mathbb{I}(h' = h).$

3. $\mathbb{P}(\tilde{\mathbf{u}}_i \mid \tilde{z}_i = (h', j), \tilde{\mathbf{x}}_j = \boldsymbol{x}_j) = \mathcal{N}(\tilde{\mathbf{u}}_i | \mathbf{W}_{O,h'}\mathbf{W}_{V,h'}\boldsymbol{x}_j, \sigma^2\mathbf{I})$. where $\sigma > 0$ is a standard deviation scalar.

4. The probability of selecting expert $k$ for token $i$, given its embedding $\boldsymbol{u}_i$, is determined by a softmax function: $\mathbb{P}(\tilde{e}_i = k \mid \tilde{\mathbf{u}}_i = \boldsymbol{u}_i) = \text{softmax}(\boldsymbol{u}_i^\top \mathbf{W}[k] + \boldsymbol{b}[k])$, where $\mathbf{W}, \boldsymbol{b}$ are defined in Section 1.2

5. Unlike the graph $\mathcal{G}_2$, under the graph $\mathcal{G}_3$, $\tilde{d}_i$ is no longer conditional independent of $\tilde{\mathbf{X}}$ given $\tilde{\mathbf{U}}$ i.e., $\mathbb{P}(\tilde{d}_i = k \mid \tilde{\mathbf{e}} = \mathbf{e}, \tilde{z}_i = (h', j)) = \mathbb{I}(k = e_j)$.

6. The final expert decision for token $i$ is then defined as:

$$\mathbb{P}(\tilde{d}_i = k \mid \tilde{\mathbf{e}} = \mathbf{e}, \tilde{s}_i = j) = \mathbb{I}(k = e_j).$$

Here, $\tilde{\mathbf{e}} = [\tilde{e}_1, \ldots, \tilde{e}_n]^T$ represents the choice of expert for all token for token $i$ with $e_i$ as its realization. Equation 6 implies that similar tokens are more likely to be routed to the same expert, promoting consistency in the processing of related information.

7. $\mathbb{P}(\tilde{\mathbf{o}}_i \mid \tilde{\mathbf{u}}_i = \boldsymbol{u}_i, \tilde{d}_i = k) = \mathcal{N}(\tilde{\mathbf{o}}_i \mid \mathbf{g}_k(\boldsymbol{u}_i), \mathbf{I})$.

**Proof**: Following the above generative process, starting with the probability of final decision for token $i$ given the sequence $\tilde{\mathbf{U}}$ and $\tilde{\mathbf{X}}$:

$$\mathbb{P}(\tilde{d}_i = k \mid \tilde{\mathbf{U}}, \tilde{\mathbf{X}}) = \sum_{h=1}^{H} \sum_{h'=1}^{H} \sum_{j=1}^{N} \sum_{e_1=1}^{K} \cdots \sum_{e_N=1}^{K} \mathbb{P}(\tilde{d}_i = k \mid \tilde{\mathbf{e}} = \mathbf{e}, \tilde{z}_i = (h', j))$$

$$\times \mathbb{P}(\tilde{z}_i = (h', j) \mid \tilde{h}_i = h, \tilde{\mathbf{u}}_i, \tilde{\mathbf{X}}) \mathbb{P}(\tilde{h}_i = h \mid \tilde{\mathbf{u}}_i, \tilde{\mathbf{X}}) \prod_{i'=1}^{N} \mathbb{P}(\tilde{e}_{i'} = e_{i'} \mid \tilde{\mathbf{u}}_i), \tag{19}$$

where $\mathbb{P}(\tilde{\mathbf{e}} = \mathbf{e} \mid \tilde{\mathbf{U}}, \tilde{\mathbf{X}}) = \prod_{i'=1}^{N} \mathbb{P}(\tilde{e}_{i'} = e_{i'} \mid \tilde{\mathbf{u}}_i)$ since $\tilde{e}_i$, for $i = 1, \ldots, N$ is conditionally mutually independent and conditionally independent on $\tilde{\mathbf{X}}$ given $\tilde{\mathbf{U}}$. Equation (19) computes the mutual-inform expert decision probability by marginalizing over all possible over head selections, attention positions, and original expert assignments.

By combining terms and using the expert assignment indicator from step 5, the RHS of Equation (19) becomes:

$$\sum_{h=1}^{H} \sum_{h'=1}^{H} \sum_{j=1}^{N} \mathbb{P}(\tilde{e}_j = k \mid \tilde{\mathbf{u}}_j) \mathbb{P}(\tilde{z}_i = (h', j) \mid \tilde{h}_i = h, \tilde{\mathbf{u}}_i, \mathbf{X}) \mathbb{P}(\tilde{h}_i = h \mid \tilde{\mathbf{u}}_i, \tilde{\mathbf{X}})$$

$$= \sum_{h=1}^{H} \sum_{j=1}^{N} \mathbb{P}(\tilde{e}_j = k \mid \tilde{\mathbf{u}}_j) \mathbb{P}(\tilde{z}_i = (h, j) \mid \tilde{h}_i = h, \tilde{\mathbf{u}}_i, \mathbf{X}) \mathbb{P}(\tilde{h}_i = h \mid \tilde{\mathbf{u}}_i, \tilde{\mathbf{X}}) \tag{20}$$

The posterior distribution of attention variable given the observation of MoE input $\tilde{\mathbf{u}}_i$ is

$$\mathbb{P}(\tilde{z}_i = (h', j) \mid \tilde{h}_i = h, \tilde{\mathbf{u}}_i, \tilde{\mathbf{X}}) = \frac{\mathbb{P}(\tilde{z}_i = (h', j) \mid \tilde{h}_i = h, \tilde{\mathbf{X}}) \mathbb{P}(\tilde{\mathbf{u}}_i \mid \tilde{z}_i = (h', j), \boldsymbol{x}_j)}{\sum_{j'} \mathbb{P}(\tilde{z}_i = (h', j)' \mid \tilde{h}_i = h, \tilde{\mathbf{X}}) \mathbb{P}(\tilde{\mathbf{u}}_i \mid \tilde{z}_i = (h', j)', \boldsymbol{x}'_j)}$$

$$= \begin{cases} \dfrac{\mathbf{A}_h[i,j] \mathcal{N}(\tilde{\mathbf{u}}_i \mid \mathbf{W}_{O,h}\mathbf{W}_{V,h}\boldsymbol{x}_j, \mathbb{I})}{\sum_{j'} \mathbf{A}_h[i,j'] \mathcal{N}(\tilde{\mathbf{u}}_i \mid \mathbf{W}_{O,h}\mathbf{W}_{V,h}\boldsymbol{x}_{j'}, \mathbb{I})} = \mathbf{A}_h^{'}[i,j], \text{if } h' = h, \\ 0, \text{ otherwise,} \end{cases}$$

$$\tag{21}$$

where $\mathbf{A}_h$ is the attention matrix of head $h$.

Then, the posterior probability of head index given input $\tilde{\mathbf{u}}_i$ of the (S)MoE and input $\tilde{\mathbf{X}}$ of the attention. This represents the responsibility of head $h$ in explaining token $i$.

$$
\begin{aligned}
\mathbb{P}(\tilde{h}_i = h \mid \tilde{\mathbf{u}}_i, \tilde{\mathbf{X}}) &= \frac{\mathbb{P}(\tilde{h}_i = h \mid \boldsymbol{x}_i) \sum_{j=1}^N \mathbb{P}(\tilde{z}_i = (h', j) \mid \tilde{h}_i = h, \tilde{\mathbf{X}}) \mathbb{P}(\tilde{\mathbf{u}}_i \mid \tilde{z}_i = (h', j), \boldsymbol{x}_j)}{\sum_{h'=1}^H \mathbb{P}(\tilde{h}_i = h' \mid \boldsymbol{x}_i) \sum_{j'} \mathbb{P}(Z_{i,h'} = j' \mid \tilde{h}_i = h', \tilde{\mathbf{X}}) \mathbb{P}(\tilde{\mathbf{u}}_i \mid Z_{i,h'} = j', \boldsymbol{x}'_j)} \\
&= \frac{\mathbf{H}[i, h] \sum_j \mathbf{A}_h[i, j] \mathcal{N}(\tilde{\mathbf{u}}_i \mid \mathbf{W}_{O,h'} \mathbf{W}_{V,h'} \boldsymbol{x}_j, \mathbb{I})}{\sum h' \mathbf{H}[i, h'] \sum_{j'} \mathbf{A}_{h'}[i, j'] \mathcal{N}(\tilde{\mathbf{u}}_i \mid \mathbf{W}_{O,h'} \mathbf{W}_{V,h'} \boldsymbol{x}_{j'}, \mathbb{I})} \\
&= \mathbf{H}'[i, h].
\end{aligned}
\tag{22}
$$

Thus, we have derived the complete dependency of $\tilde{d}_i$ on $\tilde{\mathbf{U}}$ and $\mathbf{X}$ through the expert selection process of Attention-Inform SMoE, proving Lemma 1.

## A.2 PROOF OF PROPOSITION 1

**Restate Proposition 1**

**Proposition 1.** *Let* $\mathbf{p}_i = [p_1, \ldots, p_K]^T$ *be the distribution of the final decision variable* $\tilde{d}_i$, *representing the final routing score of token* $i$. *Whereas the original routing score of token* $i$, *as defined in Definition 1, is denoted as* $\bar{\mathbf{e}}_i$. *Applying Mutual-Inform MoE to recalculate the tokens' decision score yields* $\mathbf{p}_i = \sum_{j=1}^{|J_i|} s(i, j) \bar{\mathbf{e}}_j$. *Thus, the upper bound of entropy of the final decision is given by:*

$$
\mathcal{H}(\mathbf{p}_i) \leq \sum_{j=1}^{|J_i|} s(i, j) \mathcal{H}(\bar{\mathbf{e}}_j) + \mathcal{H}(\mathbf{s}_i).
\tag{23}
$$

*where* $\mathbf{s}_i = [s(i, 1), \ldots, s(i, |J_i|)]^T$. *And as* $\tau \to 0$ *(for Similarity-Inform) or* $\sigma \to 0$ *(for Attention-Inform),* $\mathcal{H}(\mathbf{p}_i) \leq \mathcal{H}(\bar{\mathbf{e}}_i)$.

**Proof:** From $\mathbf{p}_i = \sum_{j=1}^{|J_i|} s(i, j) \bar{\mathbf{e}}_j$, omitting dependencies of $\tilde{d}_i$ and $\tilde{e}_j$ for convenience, we have $\tilde{d}_i$ is the mixture of $|J_i|$ discrete distribution of $\tilde{e}_j$ with the probability mass $\bar{\mathbf{e}}_i$. Denote $\tilde{t}_i$ is the latent random variable of that admit the weighting coefficient as probability distribution. We obtain the decomposition of joint entropy as follow

$$
\mathcal{H}(\tilde{d}_i, \tilde{t}_i) = \mathcal{H}(\tilde{t}_i) + \mathcal{H}(\tilde{d}_i \mid \tilde{t}_i) = \mathcal{H}(\tilde{t}_i) + \sum_{j=1}^{|J_i|} s(i, j) \mathcal{H}(\tilde{e}_j)
$$

Since entropy is non-negative,

$$
\mathcal{H}(\tilde{d}_i, \tilde{t}_i) = \mathcal{H}(\tilde{d}_i) + \mathcal{H}(\tilde{t}_i \mid \tilde{d}_i) \geq \mathcal{H}(\tilde{d}_i)
$$

Hence,

$$
\mathcal{H}(\tilde{d}_i) \leq \mathcal{H}(\tilde{t}_i) + \sum_{j=1}^{|J_i|} s(i, j) \mathcal{H}(\tilde{e}_j) \leq \mathcal{H}(\tilde{t}_i) + \mathcal{H}(\tilde{e}_i)
$$

because for any $j \in J_i$, $\mathcal{H}(\tilde{e}_i) > \mathcal{H}(\tilde{e}_j)$. Therefore, when $\tau \to 0$ or $\sigma \to 0$, $\mathcal{H}(\tilde{t}_i) \to 0$, and $\mathcal{H}(\tilde{d}_i) \leq \mathcal{H}(\tilde{e}_i)$ or $\mathcal{H}(\mathbf{p}_i) \leq \mathcal{H}(\bar{\mathbf{e}}_i)$. Again, here, we slightly abuse the notation of entropy $\mathcal{H}$, using it interchangeably for both a random variable and its associated distribution.

The final piece of this Proposition's proof is to verify the above limit. For $\tau \to 0$, the temperature-softmax distribution gradually morphs into an one-hot distribution, and thus its entropy goes to 0. Similarly for $\sigma \to 0$, $\mathbb{P}(\tilde{\mathbf{u}}_i \mid \tilde{z}_i = (h', j), \tilde{\mathbf{x}}_j = \boldsymbol{x}_j) = \mathcal{N}(\tilde{\mathbf{u}}_i \mid \mathbf{W}_{O,h'} \mathbf{W}_{V,h'} \boldsymbol{x}_j, \sigma^2 \mathbf{I})$ also converges to the Dirac delta function centered at the mean. This means that the closest mean will give a density greatly dominating the others, in turn making $\boldsymbol{A}'_{h^*}$ the one-hot distribution, yielding zero entropy.

With that, we have proved Proposition 1.

### A.3 SMoE Equivalence and its proof

**Renormalization.** We define the normalization of top M operator as

$$\text{TopM\_Renormalize}(\bar{\mathbf{r}})[j] := \frac{\text{TopM}(\bar{\mathbf{r}})[j]}{\sum_{k=1}^{K} \text{TopM}(\bar{\mathbf{r}})[k]}.$$

We obtain the equivalent of (2):

$$\bar{\mathbf{o}}_i = \sum_{k=1}^{K} \text{TopM\_Renormalize}(\bar{\mathbf{r}}(\bar{\mathbf{u}}_i))[k] \mathbf{g}_k(\bar{\mathbf{u}}_i) \tag{24}$$

This linear coefficient calculation process is equivalence to an alternative implementations of SMoE, which calculates the softmax probability in Eq. 2 *before* selecting the Top-M, $\bar{\mathbf{r}}(\bar{\mathbf{u}}_i) = [\text{softmax}(r_1(\bar{\mathbf{u}}_i)), \dots, \text{softmax}(r_K(\bar{\mathbf{u}}_i))]^{\top} = [\bar{r}_1, \dots, \bar{r}_K]^{\top}$, which then gets renormalized to become a proper distribution.

**Proof:** We want to show that (25) is equivalent of (2), which we restate below for further clarity:

$$\bar{\mathbf{o}}_i = \sum_{k=1}^{K} \text{softmax}(\text{TopM}(\mathbf{r}(\bar{\mathbf{u}}_i))[k]) \mathbf{g}_k(\bar{\mathbf{u}}_i),$$

Let $\mathbf{j}$ be a permutation of $[n]$ such that $r_{j_k} \geq r_{j_l}$ for all $k > l$; that is, $\mathbf{j}$ is a reordering of $\mathbf{r}$ in decreasing order. Since exponentiation is an increasing function, the post-softmax components retain the same decreasing order; that is:

$$\frac{\exp(r_{j_k})}{\sum_{i=1}^{K} \exp(r_i)} \geq \frac{\exp(r_{j_l})}{\sum_{i=1}^{K} \exp(r_i)}$$

for all $k > l$. We now prove that Top-M before softmax is equivalent to TopM_Renormalize, divided into two cases:

- If $k \leq M$, we have:

$$\text{softmax}(\text{TopM}(\mathbf{r}))_{j_k} = \frac{\exp(r_{j_k})}{\sum_{l=1}^{M} \exp(r_{j_l}) + \sum_{l=M+1}^{K} \exp(-\infty)}$$

$$= \frac{\exp(r_{j_k})}{\sum_{l=1}^{M} \exp(r_{j_l})}$$

$$= \frac{\exp(r_{j_k})}{\sum_{m=1}^{K} \exp(r_m)} \Big/ \frac{\sum_{l=1}^{M} \exp(r_{j_l})}{\sum_{m=1}^{K} \exp(r_m)}$$

$$= \text{softmax}(\mathbf{r})_{j_k} \Big/ \sum_{l=1}^{M} \text{softmax}(\mathbf{r})_{j_l}$$

$$= \text{TopM\_Renormalize}(\text{softmax}(\mathbf{r}))_{j_k}.$$

- Similarly, if $k > M$, we get $\text{softmax}(\text{TopM}(\mathbf{r}))_{j_k} = \text{TopM\_Renormalize}(\text{softmax}(\mathbf{r}))_{j_k} = 0$.

As we covered all possible values of $k$, we thus concludes our proof.

## B DERIVATION

**Optimal Regression Function.**

$$\inf_{\boldsymbol{f}} \mathbb{E}[L(\tilde{\mathbf{X}}, \tilde{\mathbf{O}})] = \inf_{\boldsymbol{f}} \int \|\boldsymbol{f}(\mathbf{X}) - \mathbf{O}\|_F^2 \, p(\mathbf{X}, \mathbf{O}) \, d\boldsymbol{x}_1 \dots d\boldsymbol{x}_N \, d\boldsymbol{o}_1 \dots d\boldsymbol{o}_N,$$

where $p(\mathbf{X}, \mathbf{O})$ is a joint density of the distribution of $\mathbf{X}$ and $\mathbf{O}$. We solve the optimization by setting the gradient of $\mathbb{E}[L(\tilde{\mathbf{X}}, \tilde{\mathbf{O}})]$ w.r.t $\boldsymbol{f}(\mathbf{X})$ to 0, then find the root of the equation: $\nabla_{\boldsymbol{f}(\mathbf{X})} \mathbb{E}[L(\tilde{\mathbf{X}}, \tilde{\mathbf{O}})] =$

$2 \int \left( \boldsymbol{f}(\mathbf{X}) - \mathbf{O} \right) p(\mathbf{X}, \mathbf{O}) \, d\boldsymbol{o}_1 \ldots d\boldsymbol{o}_N = 0$. We get

$$\boldsymbol{f}^\star(\mathbf{X}) = \frac{\int \mathbf{U} p(\mathbf{X}, \mathbf{O}) d\boldsymbol{o}_1 \ldots d\boldsymbol{o}_N}{p(\mathbf{X})}$$

$$= \mathbb{E}[\tilde{\mathbf{O}} \mid \tilde{\mathbf{X}} = \mathbf{X}] = \left[ \mathbb{E}[\tilde{\boldsymbol{o}}_1 \mid \mathbf{X} = \mathbf{X}], \ldots, \mathbb{E}[\tilde{\boldsymbol{o}}_N \mid \mathbf{X} = \mathbf{X}] \right].$$

**Multihead Attention.**

$$\mathbb{E}[\tilde{\mathbf{u}}_i \mid \tilde{\mathbf{X}}] = \mathbb{E}\left[ \mathbb{E}\left[ \mathbb{E}[\tilde{\mathbf{u}}_i \mid \tilde{z}_i, \tilde{h}_i, \tilde{\mathbf{X}}] \mid \tilde{h}_i, \tilde{\mathbf{X}}\right] \mid \tilde{\mathbf{X}} \right]$$

$$= \sum_h^H \sum_j^N \mathbb{P}(\tilde{z}_i = (h, j) \mid \tilde{h}_i = h, \mathbf{X}) \mathbb{P}(\tilde{h}_i = h \mid \boldsymbol{x}_i) \mathbb{E}[\tilde{\mathbf{u}}_i \mid \tilde{z}_i = (h, j), \boldsymbol{x}_j]$$

$$= \frac{1}{H} \sum_{h=1}^H \mathbf{W}_{O,h} \sum_{j=1}^N \mathrm{softmax}\left( \frac{\mathbf{q}_{i,h}^\top \mathbf{k}_{j,h}}{\sqrt{D_{qk}}} \right) \mathbf{v}_{j,h}.$$

**MoE Transformer Block.**

$$\mathbb{E}[\tilde{\mathbf{o}}_i \mid \mathbf{X}] = \mathbb{E}\left[ \mathbb{E}[\mathbb{E}[\tilde{\mathbf{o}}_i \mid \tilde{e}_i, \tilde{\mathbf{u}}_i] \mid \tilde{\mathbf{u}}_i] \mid \tilde{\mathbf{X}} \right]$$

$$= \mathbb{E}\left[ \sum_k^K \mathrm{softmax}(\boldsymbol{u}_i^\top \mathbf{W}_k) \mathbb{E}(\mathbf{O}_i \mid \tilde{\mathbf{u}}_i = \boldsymbol{u}_i, \tilde{e}_i = k) \mid \tilde{\mathbf{X}} \right]$$

$$= \mathbb{E}\left[ \sum_{k=1}^K \mathrm{softmax}(\boldsymbol{u}_i^\top \mathbf{W}_k) \mathbf{g}_k(\boldsymbol{u}_i) \mid \tilde{\mathbf{X}} \right].$$

## C Experiments Details

### C.1 WikiText-103 Language Modeling

**Dataset:** The WikiText-103 dataset , sourced from Wikipedia, is crafted to examine extended contextual relationships. Its training component encompasses roughly 28,000 articles, totaling 103 million words. These articles are segmented into blocks of about 3,600 words each. The validation and test sets consist of 60 articles each, with word counts of 218,000 and 246,000 respectively, amounting to approximately 268,000 words combined. To assess the resilience of our methods, we employ TextAttack's word swap attack to modify both the validation and test data. This adversarial method randomly substitutes words with "AAA," challenging the model's ability to accurately predict subsequent words in the sequence.

**Models and baselines:** In our study, we utilize the Switch Transformer (denoted as SMoE in our data presentations) and GLaM as baseline models. The Switch Transformer substitutes all multilayer perceptron (MLP) layers with SMoE layers, while GLaM replaces every alternate MLP layer. Our standard model for experiments is medium-sized with 6 layers. Each model incorporates 16 experts in every models, selecting Top-1 or Top-2 experts (K = 2) per input. All models employ an identical sparse router function, comprising a linear network that processes input data, followed by TopK and Softmax functions. The models undergo 60 epochs of training, while GLaM models train for 80 epochs without any additional load balancing loss. Our implementation builds upon the codebase developed by , which is publicly accessible at https://github.com/ofirpress/sandwich_transformer and https://github.com/giangdip2410/CompeteSMoE/tree/main.

The SMoE baseline-medium size models contains 6 layers, and 215M parameters with model sizes of 352. Whereas that config for SMoE baseline-large size and GLAM are (12 layers, 388M, model size = 512) and (6 layers, 201M, model size = 352 ) respectively

In all our Mutual-Inform SMoEs, we set the hyperparameter $\tau = 1$. In Similarity-Inform SMoE, instead of learning $\mathbf{W}_s$ in (5), we set $\mathbf{W}_s = \mathbf{I}$ for the save of computation and to avoid introduce extra parameters. In Attention-Inform SMoE, we set the hyperparameter $\sigma = 1$.

972
973
974
975
976
977
978
979
980
981
982
983
984
985

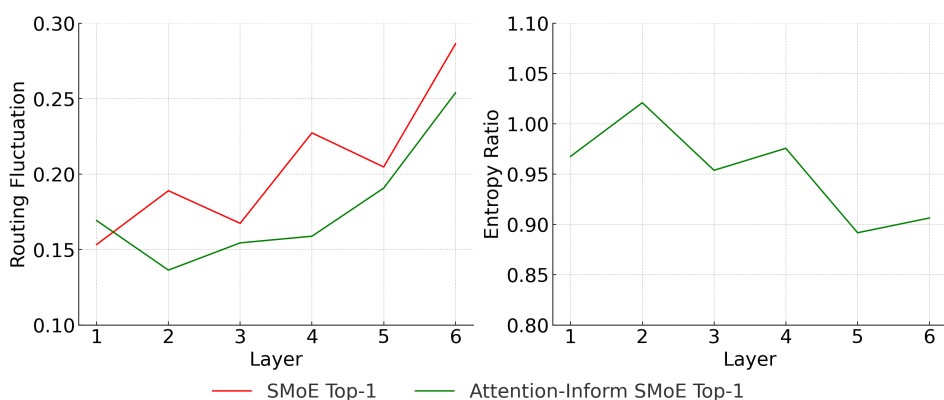

986
987
988
989

Figure 3: Comparison of Routing Fluctuation and Entropy RatioAcross Layers for Baseline SmoE Top-1, Attention-Inform SMoE Top-1, and Similarity-Inform SMoE Top-1

990
991

### C.2 IMAGENET-1K OBJECT RECOGNITION

992
993
994
995
996

**Datasets**: Our study employs the ImageNet-1K dataset, which consists of 1.28 million training images and 50,000 validation images across 1,000 object classes. The model is trained for object recognition. To evaluate resilience to input data distribution shifts, we use ImageNet-A (IN-A) . This dataset includes adversarially filtered images from a 200-class subset of ImageNet-1K. We also test our model's ability to generalize to abstract visual representations using ImageNet-R (IN-R) , which contains various artistic renditions of images.

997
998
999
1000
1001
1002
1003
1004
1005

**Model and baselines**: For our ImageNet-1K object recognition task and standard robustness benchmarks, we employ a small Vision Mixture of Experts (V-MoE) model as the SMoE baseline. This V-MoE variant is composed of 8 Vision Transformer (ViT) blocks, with the MLPs in the final two blocks replaced by SMoE layers. In our Mutual-Inform SMoEs, we alternate between Attention-Inform SMoE and Similarity-Inform SMoE layers, replacing every other MLP layer. All our vision SMoE models select 2 experts ($M = 2$) per patch at each SMoE layer. We adhere to the training configurations and settings outlined in the cited work. The codebase for this implementation is publicly available at https://github.com/google-research/vmoe/. Similar to the experiments on Language Modeling, we also we set the hyperparameter $\tau = 1$ and $\mathbf{W}_s = \mathbf{I}$ in Similarity-Inform SMoE.

1006

The VMoE baseline has 8 layers, with model size is 512 and 60M parameters.

1007
1008
1009

### D ADDITIONAL EXPERIMENTS AND ANALYSIS

1010
1011

### D.1 ROUTING FLUCTUATION AND ENTROPY OF SMOES TOP-1

1012
1013
1014
1015
1016
1017
1018
1019
1020
1021
1022

**Attention-inform SMoE Top-1 reduces routing fluctuation** Figure 3 (Left) compares the routing fluctuation of the baseline SMoE Top-1 and Attention-Inform SMoE for Top-1 routing. The fluctuation rate, computed as the proportion of tokens that switch their expert choice between consecutive last training epochs (from epoch 59 to 60), provides insight into routing stability. The Baseline SMoE exhibits higher fluctuation rates across all layers. In contrast, the Attention-Inform SMoE demonstrates consistently lower fluctuation rates across all layers. The Attention-Inform SMoE maintains more stable routing decisions throughout the network, indicating improved consistency in expert utilization. These results suggest that our proposed Attention-Inform method significantly enhances routing stability compared to the baseline approach, potentially leading to more consistent and efficient utilization of experts in the Mixture of Experts model. The results also aligns with the better performance and enhancement in robustness of Attention-Inform SMoE Top-1 in Table 1.

1023
1024
1025

**Attention-inform SMoE Top-1 reduces decision entropy** Figure 3 (Right) illustrates the ratio of average entropy of tokens' routing decisions across layers for the Attention-Inform SMoE compared to the baseline SMoE for epoch 59. The Attention-Inform SMoE demonstrates consistently lower entropy levels compared to the baseline SMoE across all layers, as evidenced by ratios below 1.0.

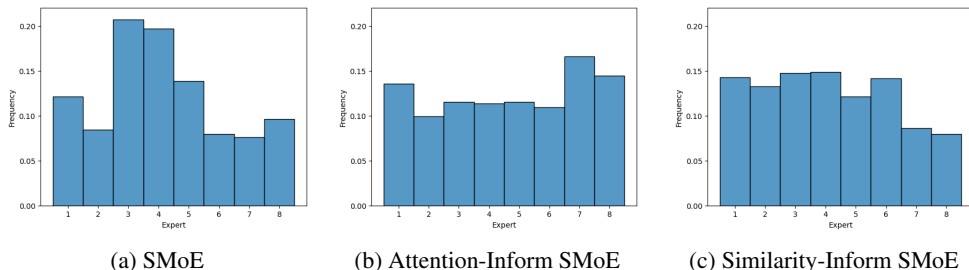

(a) SMoE  (b) Attention-Inform SMoE  (c) Similarity-Inform SMoE

Figure 4: Comparison of expert routing distribution for Baseline SMoE, Attention-Inform SMoE, and Similarity-Infom SMoE

Table 3: PPL evaluation (lower is better) with the clean and attacked Wikitext-103 on valid set and test set of SMoE-medium size variants, M=2

| Model/Metric | Clean Wikitext-103 | | Attacked Wikitext-103 | |
|---|---|---|---|---|
| | Valid PPL | Test PPL | Valid PPL | Test PPL |
| *SMoE (M = 2)* | 33.29 | 34.84 | 41.75 | 43.59 |
| Similarity-inform SMoE (M = 2) | **30.75** | **32.03** | **38.33** | **39.92** |
| Attention-inform SMoE (M = 2) | 31.31 | 32.23 | 39.68 | 40.91 |
| X-MoE (M=2) | 33.05 | 34.49 | 41.68 | 42.96 |
| Similarity-inform X-MoE | **31.83** | **33.06** | **39.92** | **41.28** |
| Attention-inform X-MoE | 32.06 | 33.24 | 40.35 | 41.73 |
| SMoE-dropout | 33.08 | 34.67 | 41.11 | 43.09 |
| Similarity-inform SMoE-dropout | 32.47 | **33.69** | 40.6 | **41.99** |
| Attention-inform SMoE-dropout | **32.21** | 33.91 | **40.56** | 42.17 |

This trend aligns with the lower routing fluctuation observed in the left graph, suggesting that our approach leads to more stable and consistent routing decisions.

### D.2 MUTUAL-INFORM SMOE ALLEVIATES LOAD IMBALANCE.

Figure 4 plots the distribution of token across experts on the VMoE architecture when we run the ImageNet test set through our model variants. As we can see from the histograms, for the baseline model, expert 3 and 4 have to take in noticeably more tokens than others. In contrast, our Mutual-Inform SMoE models spread out tokens much more evenly across experts, resembling a uniform distribution. By implicitly inducing load balancing, as input tokens move from busier experts to others, we reduce the breath of information the former experts have to learn, giving them capacity to be more specific; and prevent the freer experts from not having to learn much, as they now have to handle a wider range of input tokens.

### D.3 COMPARISON WITH PREVIOUS WORKS

To further investigate the advantages of our Mutual-inform SMoE, we compare and adapt our proposed models to X-MoE, previous work that addresses routing fluctuation in MoE models, and SMoE-dropout, another method that improves upon the standard SMoE. While both X-MoE and SMoE-dropout show improved performance over the standard SMoE baseline, as shown in Table 3, our Mutual-inform SMoE variants still significantly outperform them, as evidenced by the lower PPL scores across both Clean and Attacked Wikitext-103 datasets. Furthermore, when we integrate our proposed methods with these models to create Similarity-inform X-MoE, Attention-inform X-MoE, Similarity-inform SMoE-dropout, and Attention-inform SMoE-dropout variants, we observe substantial improvements over their respective baselines, with consistent PPL reductions in both validation and test sets. These results demonstrate not only the superior performance of our approach but also its effectiveness as a plug-and-play solution that can enhance various MoE architectures and improve model robustness.

Table 4: Top-1 test accuracy on Stanford Sentiment Treebank 5, 2 (SST5, SST2), and Banking-77 (B77) finetuning task.

| Model | sst5 | sst2 | banking77 |
|---|---|---|---|
| *SMoE* | 36.54 | 70.23 | 83.96 |
| *Similarity-inform SMoE* | 37.91 | 71.72 | 85.19 |
| *Attention-inform SMoE* | **38.89** | **72.41** | **85.84** |

Table 5: PPL evaluation (lower is better) with the clean and attacked Wikitext-103 test set Baseline SMoE (large size), Attention-Inform SMoE (large size), and Similarity-Inform SMoE (large size)

| Model/Metric | Clean Wikitext-103 | | Attacked Wikitext-103 | |
|---|---|---|---|---|
| | Valid PPL | Test PPL | Valid PPL | Test PPL |
| *SMoE (M = 2)* | 28.737 | 30.378 | 36.43 | 38.34 |
| Similarity-inform SMoE (M = 2) | **27.06** | **28.34** | **34.65** | **36.28** |
| Attention-inform SMoE (M = 2) | 27.26 | 28.69 | 34.69 | 36.37 |

## D.4 FINETUNING ON DOWNSTREAM TASKS

Regarding the adaptivity of the proposed SMoEs, we show the performance of the pretrained SMoE, the pretrained Similarity-inform SMoE, and the pretrained Attention-inform SMoE in fine-tuning. In particular, we report the test accuracy on Stanford Sentiment Treebank 5, 2 (SST5, SST2), and Banking-77 (B77) in Table 4. From the table, we observe that Attention-inform SMoE leads to the highest accuracy for all datasets. Moreover, Similarity-inform SMoE also yields better accuracy than the conventional SMoE. Overall, the result suggests that our proposed Similarity-inform SMoE and Attention-inform SMoE have better adaptivity compared to the conventional SMoEs.

## D.5 SCALABILITY OF MUTUAL-INFORM SMOES

We compare SMoE, the proposed Similarity-inform SMoE, and the proposed Mutual-inform SMoE with a large model size (about 390 million parameters) in Table 5. We observe that the scaling law happens i.e., all models perform better in language modeling when having more parameters. Moreover, we still observe that Similarity-inform SMoE and Attention-inform SMoE lead to better results than the conventional SMoE. Among all three methods, Similarity-inform SMoE is the best method.

## D.6 EXPERIMENTS WITH CHANGE IN NUMBER OF EXPERTS AND TOP-M

To evaluate performance across different model configurations, we experiment with varying numbers of experts (16 vs 32) and active experts (top-1, top-2 vs top-8). Across all these settings, both Similarity-inform SMoE and Attention-inform SMoE consistently demonstrate better performance compared to the baseline SMoE, achieving lower PPL scores on both Clean and Attacked Wikitext-103 datasets (Table 6. When using 32 experts, our methods achieve PPL reductions of up to 1.56 PPL compared to the baseline, and when increasing to top-8 active experts, they maintain their advantage with improvements of up to 1.64 PPL. These consistent performance gains across different architectural configurations demonstrate the robustness and effectiveness of our proposed methods regardless of the underlying model configuration.

## D.7 COMUPUTATION AND MEMORY

We compare the computational complexity and memory complexity of using mutual inform techniques compared to the conventional approach without them. In particular, we measure the computational time and computational memory of Similarity-Inform SMoE and Attention-Inform SMoE divided by the corresponding computational time and computational memory of the conventional SMoE in Table 7. Similarly, we report the ratio for the case of XMoE and SMoE-dropout in Table 7. From the table, we can see that mutual-inform variants only increase the computational complexities slightly.

Table 6: PPL evaluation (lower is better) with the clean and attacked Wikitext-103 test set of baseline SMoEs and Mutual-Inform SMoE(s) with different number of experts and Top-M

| Model/Metric | Clean Wikitext-103 | | Attacked Wikitext-103 | |
|---|---|---|---|---|
| | Valid PPL | Test PPL | Valid PPL | Test PPL |
| *SMoE (M = 1, K = 16)* | 39.55 | 40.75 | 48.82 | 50.21 |
| Similarity-inform SMoE (M = 1, K = 16) | 37.78 | 39.18 | 46.93 | 48.66 |
| Attention-inform SMoE (M = 1, K = 16) | 38.02 | 39.35 | 47.20 | 48.72 |
| *SMoE (M = 2, K = 16)* | 33.29 | 34.84 | 41.75 | 43.59 |
| Similarity-inform SMoE (M = 2, K = 16) | **30.75** | **32.03** | **38.33** | **39.92** |
| Attention-inform SMoE (M = 2, K = 16) | 31.31 | 32.23 | 39.68 | 40.91 |
| *SMoE (M = 8, K = 16)* | 33.48 | 34.92 | 41.36 | 42.98 |
| Similarity-inform SMoE (M = 8, K = 16) | 32.5 | 33.81 | 40.6 | 42.37 |
| Attention-inform SMoE (M = 8, K = 16) | **31.97** | **33.28** | **39.98** | **41.45** |
| *SMoE (M = 2, K = 32)* | 31.82 | 33.41 | 39.9 | 41.79 |
| Similarity-inform SMoE (M = 2,, K = 32) | 30.41 | **31.62** | 38.23 | 39.77 |
| Attention-inform SMoE (M = 2, K = 32) | **30.39** | 31.85 | **37.8** | **39.65** |

Table 7: Computation and Memory Ratio of forward pass (compared to the baselines SMoE, XMoE and SMoE-dropout) comparison for different SMoE-medium size variants, Top-M = 2

| Model | Computation Ratio | Memory Ratio |
|---|---|---|
| Similarity-Inform SMoE | 1.048 | 1.008 |
| Attention-Inform SMoE | 1.070 | 1.060 |
| Similarity-Inform XMoE | 1.026 | 1.009 |
| Attention-Inform XMoE | 1.038 | 1.060 |
| Similarity-Inform SMoE-dropout | 1.047 | 1.008 |
| Attention-Inform SMoE-dropout | 1.064 | 1.060 |

Table 8: Perplexity comparison for different SMoE variants with various $\tau$ values on validation and test sets

| Model | Valid PPL | Test PPL |
|---|---|---|
| SMoE | 33.29 | 34.84 |
| Similarity-SMoE ($\tau$=0.1) | 32.79 | 34.01 |
| Similarity-SMoE ($\tau$=1.0) | 30.75 | 32.03 |
| Similarity-SMoE ($\tau$=2.0) | 30.68 | 32.88 |
| Similarity-SMoE ($\tau$=$\sqrt{352}$) | 32.26 | 33.83 |
| Attention-SMoE ($\sigma$=0.1) | 31.93 | 32.67 |
| Attention-SMoE ($\sigma$=1.0) | 31.31 | 32.23 |
| Attention-SMoE ($\sigma$=2.0) | 31.13 | 32.85 |
| Attention-SMoE ($\sigma$=$\sqrt{352}$) | 31.62 | 32.90 |

## D.8 HYPERPARAMETER ABLATION

We present the ablation study for the hyperparameters temperatures $\tau$ in Similarity-SMoE and $\sigma$ in Attention-SMoE. Table8 demonstrates that both Similarity-SMoE and Attention-SMoE are relatively insensitive to their respective temperature parameters ($\tau$ and $\sigma$). Across different values including $0.1, 1, 2$, and $\sqrt{352}$ (where 352 is the model size). In the case of Similarity-SMoE, too large $\tau$ or too small $\tau$ can lead to an decrease in performance.

# E  ADDITIONAL MATERIALS

**Renormalization.** We define the normalization of top M operator as

$$\text{TopM\_Renormalize}(\bar{\mathbf{r}})[j] := \frac{\text{TopM}(\bar{\mathbf{r}})[j]}{\sum_{k=1}^{K} \text{TopM}(\bar{\mathbf{r}})[k]}.$$

We obtain the equivalent of (2):

$$\bar{\mathbf{o}}_i = \sum_{k=1}^{K} \text{TopM\_Renormalize}(\bar{\mathbf{r}}(\bar{\mathbf{u}}_i))[k]\mathbf{g}_k(\bar{\mathbf{u}}_i) \tag{25}$$

For a proof of equivalence, please refer to Sec. A.3.

This linear coefficient calculation process is equivalence to an alternative implementations of SMoE, which calculates the softmax probability in Eq. 2 *before* selecting the Top-M, $\bar{\mathbf{r}}(\bar{\mathbf{u}}_i) = [\text{softmax}(r_1(\bar{\mathbf{u}}_i)), \ldots, \text{softmax}(r_K(\bar{\mathbf{u}}_i))]^\top = [\bar{r}_1, \ldots, \bar{r}_K]^\top$, which then gets renormalized to become a proper distribution.

