# OpenReview forum: "Mutual-Inform SMoE: Improving Routing Stability via Probabilistic Graphical Model"
_ICLR.cc/2025/Conference — Submitted to ICLR 2025_

### Official Review · Reviewer_azAZ · 2024-11-02

**Soundness:** 4
**Presentation:** 1
**Contribution:** 3
**Rating:** 8
**Confidence:** 4

**Summary:**

This work addresses a core limitation of SMoEs—routing instability—by introducing collaborative token routing mechanisms, enhancing both model robustness and efficiency. This is show both theoretically and in a few benchmarks.

**Contributions:**

Probabilistic Graphical Model (PGM) Framework: Introduces a PGM framework to model SMoE-based attention, showing that expert selection is conditionally independent for each token, leading to unstable routing.

Mutual-Inform SMoE Models: Proposes two new SMoE variants:
- Similarity-Inform SMoE: Routes similar tokens to the same expert, using token similarities from the MoE layer.
- Attention-Inform SMoE: Utilizes relationships from the attention layer to inform expert assignments, aligning expert choice with token interactions.

This is validated theoretically (with entropy reduction) and empirically (with task specific benchmarks)

**Strengths:**

**Originality:** Introduces Mutual-Inform SMoE, a novel approach for stabilizing SMoE routing using token similarity and attention dependencies. Innovatively reinterprets SMoE through a probabilistic graphical model (PGM), which is a fresh way of addressing routing challenges.

**Quality:** Very rigorous theoretical framework (congrats!); sound mathematical proofs back claims about reduced entropy and improved stability.

**Significance:** Addresses a critical issue in SMoEs (routing instability), with potential to impact scalable model design for large-scale LLM and vision applications. Improvements in both standard and adversarial settings underscore its value in robust model deployment.

**Weaknesses:**

### Presentation weaknesses:
- Writing needs to be considerably improved (sentences unclear, leading to confusions)
- Many typos need fixing (eg: Abstract: In this work, we unveil**s**; Motivat**ing** by this PGM framework…)
- Proofs could be explained a bit better than just a series of latex equations.

### Empirical work weaknesses
- The empirical validation is weak, more benchmarks would be appreciated. The experiments could be strengthened by comparing against more recent or diverse baselines
- Mutual-Inform SMoE models show strong results, additional ablations would clarify the impact of individual components (e.g., temperature parameters, entropy reduction mechanisms). Including these would highlight the contributions of each module in performance gains.
- There is a lack of discussion on the computational costs of the method, and the computation penalty for using these more performant models compared to the baseline.

**Questions:**

- How does this work compare to other, more modern methods and across more benchmarks?
- What is the computational cost of these methods?

(See weaknesses for details)

---

> ### Author Response · Authors · 2024-11-20
> **Review reply (1/4): Q1, Q2, Q3**
>
> **Q1. Writing needs to be considerably improved (sentences unclear, leading to confusions)**
>
> **Reply:**  Thank you for your comments. We fixed all typos which are pointed out by the reviewers. We will also revise the paper based on the discussion with reviewers.
>
> **Q2 Many typos need fixing (eg: Abstract: In this work, we unveils; Motivating by this PGM framework…)**
>
> **Reply:** Thank you for identifying these typos. We have corrected all instances you pointed out and other typos found during our revision.
>
> **Q3 Proofs could be explained a bit better than just a series of latex equations.**
>
> **Reply:** Thanks for your suggestions. We will improve the proofs by adding clear explanations and intuition alongside the mathematical derivations.

---

> ### Author Response · Authors · 2024-11-20
> **Review reply (2/4): Q4**
>
> **Q4 The empirical validation is weak, more benchmarks would be appreciated. The experiments could be strengthened by comparing against more recent or diverse baselines.**
>
> **How does this work compare to other, more modern methods and across more benchmarks?**
>
> **Reply:** Thank you for your comment. To strengthen our experiment results, we have conducted the following additional experiments: 1) comparison of our method to more previous and recent works; 2) finetuning on downstream tasks; we also demonstrate 3) the scalibility and 4) benefits of our method across more settings. We present our results here.
>
> 1. To further investigate the advantages of our Mutual-inform SMoE, we ***compare and adapt*** our proposed models to X-MoE, previous work that addresses routing fluctuation in MoE models, and the more recent SMoE-dropout[2], another method that improves upon the standard SMoE. While both X-MoE[1] and SMoE-dropout show improved performance over the standard SMoE baseline, as shown in Table 1 (or Table 3, Appendix D.3), our Mutual-inform SMoE variants still significantly outperform them, as evidenced by the lower PPL scores across both Clean and Attacked Wikitext-103 datasets. Furthermore, when we integrate our proposed methods with these models to create Similarity-inform X-MoE, Attention-inform X-MoE, Similarity-inform SMoE-dropout, and Attention-inform SMoE-dropout variants, we observe substantial improvements over their respective baselines, with consistent PPL reductions in both validation and test sets. These results demonstrate not only the superior performance of our approach but also its effectiveness as a plug-and-play solution that can enhance various MoE architectures and improve model robustness.
>
> Table 1: *PPL evaluation (lower is better) with the clean and attacked Wikitext-103 on valid set and test set of SMoE-medium size variants, M=2*
> | Model/Metric | Clean Wikitext-103 Valid PPL | Clean Wikitext-103 Test PPL | Attacked Wikitext-103 Valid PPL | Attacked Wikitext-103 Test PPL |
> |------------|---------------------------|--------------------------|--------------------------------|-------------------------------|
> | SMoE | 33.29 | 34.84 | 41.75 | 43.59 |
> | Similarity-inform SMoE | **30.75** | **32.03** | **38.33** | **39.92** |
> | Attention-inform SMoE | 31.31 | 32.23 | 39.68 | 40.91 |
> | X-MoE | 33.05 | 34.49 | 41.68 | 42.96 |
> | Similarity-inform X-MoE | **31.83** | **33.06** | **39.92** | **41.28** |
> | Attention-inform X-MoE | 32.06 | 33.24 | 40.35 | 41.73 |
> | SMoE-dropout | 33.08 | 34.67 | 41.11 | 43.09 |
> | Similarity-inform SMoE-dropout | 32.47 | 33.69 | 40.6 | 41.99 |
> | Attention-inform SMoE-dropout | **32.21** | **33.91** | **40.56** | **42.17** |
>
> **References**
>
> [1]Chi et al. "On the Representation Collapse of Sparse Mixture of Experts". NeurIPS. 2022
>
> [2]Chen, et al. "*Sparse MoE as the New Dropout: Scaling Dense and Self-Slimmable Transformers*" ICLR. 2023.
>
> 2. To evaluate the ***adaptability*** of our models, we examine the fine-tuning performance of pretrained SMoE variants on different downstream tasks. Specifically, we assess their test accuracy on Stanford Sentiment Treebank 5[3], 2[4] (SST5, SST2), and Banking-77[5] (B77) datasets, as shown in Table 2 (or see Table 4, Appendix D.4). The results demonstrate that Attention-inform SMoE consistently achieves the highest accuracy across all datasets (38.89% on SST5, 72.41% on SST2, and 85.84% on B77), while Similarity-inform SMoE also consistently outperforms the conventional SMoE baseline. These improvements in fine-tuning performance across diverse tasks suggest that our proposed Mutual-inform SMoE variants possess better adaptability compared to conventional SMoEs.
>
> Table 2: *Top-1 test accuracy on Stanford Sentiment Treebank 5, 2 (SST5, SST2), and Banking-77 (B77) finetuning task for SMoE-medium size variants, M=2*
> | Model | sst5 | sst2 | banking77 |
> |------------|---------------------------|--------------------------|--------------------------------|
> | SMoE | 36.54 | 70.23 | 83.96 |
> | Similarity-inform SMoE | 37.91 | 71.72 | 85.19 |
> | Attention-inform SMoE | **38.89** | **72.41** | **85.84** |
>
> **References**
>
> [3, 4]Socher et al. "*Recursive Deep Models for Semantic Compositionality Over a Sentiment Treebank.*" EMNLP. 2013.
>
> [5]Casanueva et al. "*Efficient intent detection with dual sentence encoders." NLP4ConvAI. 2013.

---

> ### Author Response · Authors · 2024-11-20
> **Review reply (3/4): Q4**
>
> 3. To further demonstrate the ***scalability*** of our models, we evaluate them with a larger transformer-MoEs baseline of approximately 390M parameters, with 12 layers. The results in Table 3 (or Table 5 in Appendix D.5 ) confirms that the scaling law holds true, as all models show improved language modeling performance with increased parameter count. Importantly, both Similarity-inform SMoE and Attention-inform SMoE maintain their performance advantage over the conventional SMoE at this larger scale, with Similarity-inform SMoE emerging as the best performing variant. These findings validate that the benefits of our proposed methods are preserved when scaling up model size.
>
> Table 3: *PPL evaluation (lower is better) with the clean and attacked Wikitext-103 on valid set and test set of SMoE-large size variants, M=2*
> | Model/Metric | Clean Wikitext-103 Valid PPL | Clean Wikitext-103 Test PPL | Attacked Wikitext-103 Valid PPL | Attacked Wikitext-103 Test PPL |
> |------------|---------------------------|--------------------------|--------------------------------|-------------------------------|
> | SMoE | 28.737 | 30.378 | 36.43 | 38.34 |
> | Similarity-inform SMoE | **27.06** | **28.34** | **34.65** | **36.28** |
> | Attention-inform SMoE | 27.26 | 28.69 | 34.69 | 36.37 |
>
> 4. While our original submission included experiments with 16 experts and varying active experts (top-1, top-2), we have now added new experiments with 32 experts and top-2, as well as 16 experts and top-8 routing during the rebuttal period to provide a more comprehensive analysis of trends. Across all these settings, both Similarity-inform SMoE and Attention-inform SMoE consistently demonstrate better performance compared to the baseline SMoE, achieving lower PPL scores on both Clean and Attacked Wikitext-103 datasets (Table 4, or Table 6 in Appendix D.6). When using 32 experts, our methods achieve PPL reductions of up to 1.56 PPL compared to the baseline, and when increasing to top-8 active experts, they maintain their advantage with improvements of up to 1.64 PPL. These consistent performance gains across different architectural configurations demonstrate the robustness and effectiveness of our proposed methods.
>
> Table 4: *PPL evaluation (lower is better) with the clean and attacked Wikitext-103 test set of baseline SMoEs and Mutual-Inform SMoE(s) with different number of experts and Top-M.*
> | Model/Metric | Clean Wikitext-103 Valid PPL | Clean Wikitext-103 Test PPL | Attacked Wikitext-103 Valid PPL | Attacked Wikitext-103 Test PPL |
> |------------|---------------------------|--------------------------|--------------------------------|-------------------------------|
> | SMoE (M = 1, K = 16) | 39.55 | 40.75 | 48.82 | 50.21 |
> | Similarity-inform SMoE (M = 1, K = 16) | 37.78 | 39.18 | 46.93 | 48.66 |
> | Attention-inform SMoE (M = 1, K = 16) | 38.02 | 39.35 | 47.20 | 48.72 |
> | SMoE (M = 2, K = 16) | 33.29 | 34.84 | 41.75 | 43.59 |
> | Similarity-inform SMoE (M = 2, K = 16) | **30.75** | **32.03** | **38.33** | **39.92** |
> | Attention-inform SMoE (M = 2, K = 16) | 31.31 | 32.23 | 39.68 | 40.91 |
> | SMoE (M = 8, K = 16) | 33.48 | 34.92 | 41.36 | 42.98 |
> | Similarity-inform SMoE (M = 8, K = 16) | 32.5 | 33.81 | 40.6 | 42.37 |
> | Attention-inform SMoE (M = 8, K = 16) | 31.97 | **33.28** | 39.98 | **41.45** |
> | SMoE (M=2, K = 32) | 31.82 | 33.41 | **39.90** | 41.79 |
> | Similarity-inform SMoE (M=2, K = 32) | 30.41 | **31.62** | 38.23 | 39.77 |
> | Attention-inform SMoE (M=2, K = 32) | **30.39** | 31.85 | **37.8** | **39.65** |

---

> ### Author Response · Authors · 2024-11-20
> **Review reply (4/4): Q5, Q6**
>
> **Q5 Mutual-Inform SMoE models show strong results, additional ablations would clarify the impact of individual components (e.g., temperature parameters, entropy reduction mechanisms). Including these would highlight the contributions of each module in performance gains.**
>
> **Reply:** Thank you for your suggestion. We will add the an ablation study hyperparameter sensitivity in a few days.
>
> **Q6 There is a lack of discussion on the computational costs of the method, and the computation penalty for using these more performant models compared to the baseline.**
>
>
> **What is the computational cost of these methods?**
>
> **Reply:** Thanks for your suggestion. We have compared the computational and memory complexity of Mutual-Inform SMoE with those of the baseline SMoE. In particular, we compute the ratio between the run time per sample and memory usage of our Mutual-Inform SMoE models and those of the baseline SMoE, respectively. We have also attain similar ratios for XMoE and SMoE-dropout. We refer the reviewer to Table 7 in Appendix D.7 of our revision for the results, and present them in Table 5 below as well for convenience. The results confirm that our Mutual-Inform SMoE introduces only a slight and negligible increase in computational and memory complexity.
>
> Table 5: *Computation and Memory Ratio of forward pass (compared to the baselines SMoE, XMoE and SMoE-dropout) comparison for different SMoE-medium size variants, Top-M = 2*
> | Model | Computation Ratio | Memory Ratio |
> |------------|---------------------------|--------------------------|
> | Similarity-Inform SMoE | 1.048 | 1.008 |
> | Attention-Inform SMoE | 1.070 | 1.060 |
> | Similarity-Inform XMoE | 1.026 | 1.009 |
> | Attention-Inform XMoE | 1.038 | 1.060 |
> | Similarity-Inform SMoE-dropout | 1.047 | 1.008 |
> | Attention-Inform SMoE-dropout | 1.064 | 1.060 |
>
> -----
> We hope we have cleared your concerns about our work. We have also revised our manuscript according to your comments, and we would appreciate it if we can get your further feedback at your earliest convenience.

---

> > ### Author Response · Authors · 2024-11-21
> > **Review Reply: Q5**
> >
> > **Q5 Mutual-Inform SMoE models show strong results, additional ablations would clarify the impact of individual components (e.g., temperature parameters, entropy reduction mechanisms). Including these would highlight the contributions of each module in performance gains.**
> >
> > **Reply**:
> > Thank you for the reviewer's suggestions. Firstly, we have added the ablation study for hyperparameters temperatures $\tau$ in Similarity-SMoE and $\sigma$ in Attention-SMoE. Table 6 (or Table 8 Appendix D.8) below demonstrates that both Similarity-SMoE and Attention-SMoE are relatively insensitive to their respective temperature parameters ($\tau$ and $\sigma$). Across different values including $0.1, 1, 2$, and $\sqrt{352}$ (where 352 is the model size). In the case of Similarity-SMoE, too large $\tau$ or too small $\tau$ can lead to a decrease in performance.
> >
> > Secondly, in Attention-SMoE, we employ an entropy-based head selection mechanism to reduce computational overhead while maintaining model performance. Specifically, instead of computing the full posterior inference across all attention heads, we approximate it by selecting only the attention head $h^*$ that has the minimum expected entropy (Eqn. 16). The intuition is that the head with the lowest entropy exhibits the most certain/concentrated attention distribution, thus likely containing the most informative signals for the mixture assignment.
> >
> > Table 6: *PPL evaluation (lower is better) with the clean and attacked Wikitext-103 on valid set and test set of SMoE-medium size variants, M=2*
> > | Model\Metrics | Valid PPL | Test PPL |
> > |------------|-------------|------------|
> > | SMoE | 33.29 | 34.84 |
> > | Similarity-SMoE ($\tau=0.1$) | 32.79 | 34.01 |
> > | Similarity-SMoE ($\tau=1$) | 30.75 | 32.03 |
> > | Similarity-SMoE ($\tau=2$) | 30.68 | 32.88 |
> > | Similarity-SMoE ($\tau=\sqrt{352}$) | 32.26 | 33.83 |
> > | Attention-SMoE ($\sigma=0.1$) | 31.93 | 32.67 |
> > | Attention-SMoE ($\sigma=1$) | 31.31 | 32.23 |
> > | Attention-SMoE ($\sigma=2$) | 31.13 | 32.85 |
> > | Attention-SMoE ($\sigma=\sqrt{352}$) | 31.62 | 32.90 |

---

> > > ### Author Response · Authors · 2024-11-22
> > > **Any Questions from Reviewer azAZ on Our Rebuttal?**
> > >
> > > We would like to thank the reviewer again for your thoughtful reviews and valuable feedback.
> > >
> > > We would appreciate it if you could let us know if our responses have addressed your concerns and whether you still have any other questions about our rebuttal.
> > >
> > > We would be happy to do any follow-up discussion or address any additional comments.

---

### Official Review · Reviewer_Mg8s · 2024-11-03

**Soundness:** 2
**Presentation:** 2
**Contribution:** 2
**Rating:** 5
**Confidence:** 4

**Summary:**

This paper proposes a novel perspective on Sparse Mixture of Experts (SMoE) through a probabilistic graphical model (PGM) framework. The authors identify that the conditional independence in expert selection for tokens leads to routing fluctuations and model instability. To address this, they introduce Mutual-Inform SMoE with two variants: Similarity-Inform and Attention-Inform, which allow tokens to influence each other's expert assignments based on their similarities or attention patterns. They provide theoretical analysis showing their method reduces entropy in routing decisions and demonstrate empirical improvements on ImageNet classification and WikiText-103 language modeling tasks.

**Strengths:**

1. The paper provides a novel theoretical foundation for understanding SMoE through PGM, offering valuable insights into the routing fluctuation problem and presenting a well-grounded solution. The mathematical derivations and proofs are rigorous and well-presented.

2. The proposed solution is elegant and practical, requiring minimal additional computational overhead while showing significant improvements in both performance and stability. The two variants (Similarity-Inform and Attention-Inform) provide flexibility in implementation.

3. The experimental evaluation is comprehensive, covering both vision and language tasks, with thorough analysis of routing stability, entropy reduction, and robustness against various types of perturbations.

**Weaknesses:**

1. While the paper demonstrates improvements in routing stability, it doesn't fully explore the trade-offs between stability and adaptability. A more detailed analysis of whether increased stability might sometimes come at the cost of reduced model flexibility would be valuable.

2. The hyperparameter sensitivity analysis is limited, particularly regarding the temperature parameter $\tau$ in Similarity-Inform SMoE and $\sigma$ in Attention-Inform SMoE. Understanding how these parameters affect performance would be crucial for practical implementations.

3. The experiments focus primarily on medium-sized models. Given that routing issues often become more pronounced in larger-scale settings, evaluation on larger models would strengthen the paper's claims.

4. While the paper mentions load balancing benefits in the appendix, this aspect deserves more attention in the main text, as it could be a significant practical advantage of the proposed approach.

**Questions:**

1. How does the approach scale with very large numbers of experts (e.g., hundreds or thousands)?

2. Have you explored the possibility of dynamically adjusting the influence of token similarities based on training progress?

3. How does the method perform when applied to multi-modal tasks where routing patterns might be more complex?

---

> ### Author Response · Authors · 2024-11-20
> **Review Reply (1/2): Q1, Q2, Q3, Q4**
>
> **Q1 While the paper demonstrates improvements in routing stability, it doesn't fully explore the trade-offs between stability and adaptability. A more detailed analysis of whether increased stability might sometimes come at the cost of reduced model flexibility would be valuable.**
>
> **Reply:** Thank you for your comment. Based on your suggestion, we examine the fine-tuning performance of pretrained SMoE variants on different downstream tasks. Specifically, we assess their test accuracy on Stanford Sentiment Treebank 5[1], 2[2] (SST5, SST2), and Banking-77[3] (B77) datasets, as shown in Table 1 (or see Table 4, Appendix D.4). The results demonstrate that Attention-inform SMoE consistently achieves the highest accuracy across all datasets (38.89% on SST5, 72.41% on SST2, and 85.84% on B77), while Similarity-inform SMoE also consistently outperforms the conventional SMoE baseline. These improvements in fine-tuning performance across diverse tasks suggest that our proposed Mutual-inform SMoE variants possess better adaptability compared to conventional SMoEs.
>
> Table 1: *Top-1 test accuracy on Stanford Sentiment Treebank 5, 2 (SST5, SST2), and Banking-77 (B77) finetuning task for SMoE-medium size variants, M=2*
> | Model | sst5 | sst2 | banking77 |
> |------------|---------------------------|--------------------------|--------------------------------|
> | SMoE | 36.54 | 70.23 | 83.96 |
> | Similarity-inform SMoE | 37.91 | 71.72 | 85.19 |
> | Attention-inform SMoE | **38.89** | **72.41** | **85.84** |
>
> **References**
>
> [1, 2]Socher et al. "*Recursive Deep Models for Semantic Compositionality Over a Sentiment Treebank.*" EMNLP. 2013.
>
> [3]Casanueva et al. "*Efficient intent detection with dual sentence encoders.*" NLP4ConvAI. 2013.
>
> **Q2 The hyperparameter sensitivity analysis is limited, particularly regarding the temperature parameterin Similarity-Inform SMoE and in Attention-Inform SMoE. Understanding how these parameters affect performance would be crucial for practical implementations.**
>
> **Reply:** Thank you for your suggestion. We will add an ablation study hyperparameter sensitivity in a few days.
>
> **Q3 The experiments focus primarily on medium-sized models. Given that routing issues often become more pronounced in larger-scale settings, evaluation on larger models would strengthen the paper's claims.**
>
> **Reply:** To further demonstrate the scalability of our models, we evaluate them with a larger transformer-MoEs baseline of approximately 390M parameters, with 12 layers. The results in Table 2 (or Table 5 in Appendix D.5 ) confirm that the scaling law holds true, as all models show improved language modeling performance with increased parameter count. Importantly, both Similarity-inform SMoE and Attention-inform SMoE maintain their performance advantage over the conventional SMoE at this larger scale, with Similarity-inform SMoE emerging as the best performing variant. These findings validate that the benefits of our proposed methods are preserved when scaling up model size.
>
> Table 2: *PPL evaluation (lower is better) with the clean and attacked Wikitext-103 on valid set and test set of SMoE-**large** size variants, M=2*
> | Model/Metric | Clean Wikitext-103 Valid PPL | Clean Wikitext-103 Test PPL | Attacked Wikitext-103 Valid PPL | Attacked Wikitext-103 Test PPL |
> |------------|---------------------------|--------------------------|--------------------------------|-------------------------------|
> | SMoE | 28.737 | 30.378 | 36.43 | 38.34 |
> | Similarity-inform SMoE | **27.06** | **28.34** | **34.65** | **36.28** |
> | Attention-inform SMoE | 27.26 | 28.69 | 34.69 | 36.37 |
>
> **Q4 While the paper mentions load balancing benefits in the appendix, this aspect deserves more attention in the main text, as it could be a significant practical advantage of the proposed approach.**
>
> **Reply:** Thank you for the suggestion, we will move the load-balancing analysis in the maintext in our revision.

---

> ### Author Response · Authors · 2024-11-20
> **Review Reply (2/2): Q5, Q6, Q7**
>
> **Q5 How does the approach scale with very large numbers of experts (e.g., hundreds or thousands)?**
>
> **Reply:** Thank you for your interesting question. To our best knowledge, having hundreds or thousands of experts in Transformer-MoE models is not a common practice in current research. While scaling to such numbers is an interesting research direction, current hardware constraints in our lab make these experiments prohibitively expensive. Instead, we systematically analyze our method's behavior by doubling the number of experts (from 16 to 32). Table 3 below shows that when using 32 experts, our methods achieve PPL reductions of up to 1.56 PPL compared to the baseline, further demonstrate the benefit of our method. This controlled approach further provides insights into scalability while staying within computational feasibility.
>
> Table 3: *PPL evaluation (lower is better) with the clean and attacked Wikitext-103 test set of medium-size baseline SMoEs and Mutual-Inform SMoE(s) with 32 number of experts and Top-2.*
> | Model/Metric | Clean Wikitext-103 Valid PPL | Clean Wikitext-103 Test PPL | Attacked Wikitext-103 Valid PPL | Attacked Wikitext-103 Test PPL |
> |------------|---------------------------|--------------------------|--------------------------------|-------------------------------|
> | SMoE (M=2, K = 32) | 31.82 | 33.41 | 39.9 | 41.79 |
> | Similarity-inform SMoE (M=2, K = 32) | 30.41 | **31.62** | 38.23 | 39.77 |
> | Attention-inform SMoE (M=2, K = 32) | **30.39** | 31.85 | **37.8** | **39.65** |
>
> **Q6 Have you explored the possibility of dynamically adjusting the influence of token similarities based on training progress?**
>
> **Reply:** The token similarity influences decisions through the score matrix $S = softmax(UU^T/\tau)$, where $\tau$ is the temperature parameter. As $\tau \to 0$, $S$ converges to the identity matrix, meaning each token becomes conditionally independent in decision-making. We are currently investigating how gradually varying $\tau$ during training affects model performance and will report results shortly.
>
> **Q7 How does the method perform when applied to multi-modal tasks where routing patterns might be more complex?**
>
> **Reply:** Thank you very much for your insightful question. We agree that routing patterns might be more complex in multi-modal tasks and reducing routing fluctuation can lead to considerable improvements. Nevertheless, multi-modal tasks require a lot of computations and time to train and evaluate. Therefore, we are not able to conduct experiments due to the time limitation of the discussion period. Since MoEs have been widely used in many applications, exploring the full potential of the proposed mutual informed techniques is worth further investiation. Therefore, we will leave the direction on multi-modal tasks for future works. From the exisiting experiments, we believe that it is sufficient to show that mutual informed MoEs are able to enhance performance of MoEs' applications by stablizing routing.
>
> -----
> We hope we have cleared your concerns about our work. We have also revised our manuscript according to your comments, and we would appreciate it if we can get your further feedback at your earliest convenience.

---

> ### Author Response · Authors · 2024-11-21
> **Review Reply: Q2**
>
> **Q2 The hyperparameter sensitivity analysis is limited, particularly regarding the temperature parameter in Similarity-Inform SMoE and Attention-Inform SMoE. Understanding how these parameters affect performance would be crucial for practical implementations.**
>
> **Reply**: Thank you for the reviewer's suggestions. We have added the ablation study for the hyperparameters temperatures $\tau$ in Similarity-SMoE and $\sigma$ in Attention-SMoE. Table 4 below (or Table 8 in Appendix D.8) demonstrates that Similarity-SMoE and Attention-SMoE are relatively insensitive to their respective temperature parameters ($\tau$ and $\sigma$). Across different values including $0.1, 1, 2$, and $\sqrt{352}$ (where 352 is the model size). In the case of Similarity-SMoE, too large $\tau$ or too small $\tau$ can lead to a decrease in performance.
>
>
> Table: *PPL evaluation (lower is better) with the clean and attacked Wikitext-103 on valid set and test set of SMoE-medium size variants, M=2*
> | Model\Metrics | Valid PPL | Test PPL |
> |------------|-------------|------------|
> | SMoE | 33.29 | 34.84 |
> | Similarity-SMoE ($\tau=0.1$) | 32.79 | 34.01 |
> | Similarity-SMoE ($\tau=1$) | 30.75 | 32.03 |
> | Similarity-SMoE ($\tau=2$) | 30.68 | 32.88 |
> | Similarity-SMoE ($\tau=\sqrt{352}$) | 32.26 | 33.83 |
> | Attention-SMoE ($\sigma=0.1$) | 31.93 | 32.67 |
> | Attention-SMoE ($\sigma=1$) | 31.31 | 32.23 |
> | Attention-SMoE ($\sigma=2$) | 31.13 | 32.85 |
> | Attention-SMoE ($\sigma=\sqrt{352}$) | 31.62 | 32.90 |

---

> > ### Author Response · Authors · 2024-11-22
> > **Any Questions from Reviewer Mg8s on Our Rebuttal?**
> >
> > We would like to thank the reviewer again for your thoughtful reviews and valuable feedback.
> >
> > We would appreciate it if you could let us know if our responses have addressed your concerns and whether you still have any other questions about our rebuttal.
> >
> > We would be happy to do any follow-up discussion or address any additional comments.

---

> > > ### Author Response · Authors · 2024-11-23
> > > **Review Reply: Q6 (Continue)**
> > >
> > > **Q6 Have you explored the possibility of dynamically adjusting the influence of token similarities based on training progress?**
> > >
> > > **Reply:** We studied how gradually varying $\tau$ during training affects model performance. As expected, when $\tau \to 0$, the performance of Similarity-Informed SMoE becomes comparable with the baseline SMoE. On the other hand, increasing $\tau$ from 0.01 to 10 improves the performance of the baseline.
> > >
> > > Table 1: *PPL evaluation (lower is better) with the clean and attacked Wikitext-103 test set of medium-size baseline SMoEs and Similarity-Inform SMoE(s) with $\tau$ decreases from 10 to 0.01 and increases from 0.01 to 10*
> > > | Model/Metric | Clean Wikitext-103 Valid PPL | Clean Wikitext-103 Test PPL | Attacked Wikitext-103 Valid PPL | Attacked Wikitext-103 Test PPL |
> > > |------------|---------------------------|--------------------------|--------------------------------|-------------------------------|
> > > | SMoE | 33.29 | 34.84 | 41.75 | 43.59 |
> > > | Similarity-inform some ($\tau$) decreases | 33.37 | 34.65 | 41.61 | 43.08 |
> > > | Similarity-inform some ($\tau$) increases| 32.40 | 33.91 | 39.92 | 41.67|

---

### Official Review · Reviewer_jpxc · 2024-11-04

**Soundness:** 2
**Presentation:** 2
**Contribution:** 3
**Rating:** 5
**Confidence:** 4

**Summary:**

The paper proposes "MUTUAL-INFORM SMOEs", MoEs with a tweaked routing mechanism that allows for information sharing between routed tokens. The authors motivate their proposed method as a solution to reducing the routing fluctuation problem in MoEs. They provide theoretical justification for their method's reduction in routing fluctuation by showing that the entropy of routing decisions of "MUTUAL-INFORM SMOEs" is upper bounded by standard topK + softmax routing. The authors provide additional experiments on language modeling (standard + adversarially perturbed text) and image classification, showing that their method outperforms a standard topK + softmax baseline in routing fluctuation and validation perplexity.

**Strengths:**

- The Attention-Inform (S)MoE and Similarity-Inform (S)MoE routing functions are novel and potentially interesting to the community. The introduction of routing mechanisms that can share information between tokens can lead to better-performing MoEs.
- It is nice to see the theoretical justification for the proposed method.
- I like the idea of including an adversarial test set.
- Providing code for the method is appreciated and helps validate the results.

**Weaknesses:**

My **main concern** is the lack of comparison to previous work that also improves the *routing fluctuation* problem of MoEs. Authors cite [1,2] numerous times when referring to the routing fluctuation problem. However, they do not provide a comparison to Stablemoe or X-MoE. To the best of my knowledge, [1,2] are the only works that claim this *routing fluctuation* problem exists in MoEs and to provide a solution for it. As clear follow-up work to [1,2], both more than two years old, it is very important to directly compare to their method. Adding a comparison to these methods would greatly strengthen the contributions of the work.

**Other concerns**
- The method relies on the attention matrix for similarity scores between different tokens. How does this influence the computational and memory complexity of an MoE forward pass? Providing figures that show the timings with respect to a baseline would address this concern.
- The introduction is structured unconventionally and I found myself taking time to find the motivation for the work.
- It would be helpful to understand how the performance of M-I SMoE changes as the number of experts is increased, the granularity of the experts changes, and the number of active parameters changes.
- The architecture of the MoEs used (e.g., number of experts per layer) should be reported in the main text, not only in the appendix.
- I am unable to find the dimensions of the transformer used in the language modeling experiments.


Typos:
- "Motivating by this PGM framework" Motivating --> Motivated
- Page 10: "We leave it for future work. We leave it for future work"


[1] Stablemoe: Stable routing strategy for mixture of experts ( https://aclanthology.org/2022.acl-long.489/ )
[2] On the Representation Collapse of Sparse Mixture of Experts ( https://arxiv.org/pdf/2204.09179 )

**Questions:**

- Is routing fluctuation discussed as a problem for MoEs outside of papers [1,2]?


[1] Stablemoe: Stable routing strategy for mixture of experts ( https://aclanthology.org/2022.acl-long.489/ )
[2] On the Representation Collapse of Sparse Mixture of Experts ( https://arxiv.org/pdf/2204.09179 )

---

> ### Author Response · Authors · 2024-11-20
> **Review reply (1/3): Q1, Q2**
>
> **Q1 My main concern is the lack of comparison to previous work that also improves the routing fluctuation problem of MoEs. Authors cite [1,2] numerous times when referring to the routing fluctuation problem. However, they do not provide a comparison to Stablemoe or X-MoE. To the best of my knowledge, [1,2] are the only works that claim this routing fluctuation problem exists in MoEs and to provide a solution for it. As clear follow-up work to [1,2], both more than two years old, it is very important to directly compare to their method. Adding a comparison to these methods would greatly strengthen the contributions of the work.**
>
> **Reply:**  Thanks for your comments. We have conducted additional experiments comparing our Mutual-Inform SMoE models with X-MoE [1], previous work that addresses routing fluctuation in MoE models, and SMoE-dropout[2], another method that improves upon the standard SMoE, which initially randomizes and freezes the router during training to provide stable routing strategie. While both X-MoE and SMoE-dropout show improved performance over the standard SMoE baseline, as shown in Table 1 below (or Table 3, Appendix D.3), our Mutual-inform SMoE variants still significantly outperform them, as evidenced by the lower PPL scores across both clean and attacked Wikitext-103 datasets. Furthermore, when we integrate our proposed methods with X-MoE and SMoE-dropout to create Similarity-inform X-MoE, Attention-inform X-MoE, Similarity-inform SMoE-dropout, and Attention-inform SMoE-dropout variants, we observe substantial improvements over their respective baselines, with consistent PPL reductions in both validation and test sets. These results demonstrate not only the advantages of our approach but also its effectiveness as a plug-and-play solution that can enhance various MoE architectures.
>
> We have also tried to compare with StableMoE. However, due to the significant differences between the StableMoE github (https://github.com/Hunter-DDM/stablemoe) and our current code, as well as the limited time we have for the rebuttal, we decided to compare with SMoE-dropout instead. Like StableMoE, SMoE-dropout also proposes a new training strategy for the routers and experts, in which the routers are randomized and frozen.
>
> Table 1: *PPL evaluation (lower is better) with the clean and attacked Wikitext-103 on valid set and test set of SMoE-medium size variants, M=2*
> | Model/Metric | Clean Wikitext-103 Valid PPL | Clean Wikitext-103 Test PPL | Attacked Wikitext-103 Valid PPL | Attacked Wikitext-103 Test PPL |
> |------------|---------------------------|--------------------------|--------------------------------|-------------------------------|
> | SMoE | 33.29 | 34.84 | 41.75 | 43.59 |
> | Similarity-inform SMoE | **30.75** | **32.03** | **38.33** | **39.92** |
> | Attention-inform SMoE | 31.31 | 32.23 | 39.68 | 40.91 |
> | X-MoE | 33.05 | 34.49 | 41.68 | 42.96 |
> | Similarity-inform X-MoE | **31.83** | **33.06** | **39.92** | **41.28** |
> | Attention-inform X-MoE | 32.06 | 33.24 | 40.35 | 41.73 |
> | SMoE-dropout | 33.08 | 34.67 | 41.11 | 43.09 |
> | Similarity-inform SMoE-dropout | 32.47 | 33.69 | 40.6 | 41.99 |
> | Attention-inform SMoE-dropout | **32.21** | **33.91** | **40.56** | **42.17** |
>
> **References**
>
> [1]Chi et al. "*On the Representation Collapse of Sparse Mixture of Experts*". NeurIPS. 2022
>
> [2]Chen, et al. "*Sparse MoE as the New Dropout: Scaling Dense and Self-Slimmable Transformers*" ICLR. 2023.
>
> **Q2 The method relies on the attention matrix for similarity scores between different tokens. How does this influence the computational and memory complexity of an MoE forward pass? Providing figures that show the timings with respect to a baseline would address this concern.**
>
> **Reply:** Thanks for your suggestion. We have compared the computational and memory complexity of Mutual-Inform SMoE with those of the baseline SMoE. In particular, we compute the ratio between the run time per sample and memory usage of our Mutual-Inform SMoE models and those of the baseline SMoE, respectively. We have also attain similar ratios for XMoE and SMoE-dropout. We refer the reviewer to Table 7 in Appendix D.7 of our revision for the results, and present them in Table 2 below as well for convenience. The results confirm that our Mutual-Inform SMoE introduces only a slight and negligible increase in computational and memory complexity.
>
> Table 2: *Computation and Memory Ratio of forward pass (compared to the baselines SMoE, XMoE and SMoE-dropout) comparison for different SMoE-medium size variants, Top-M = 2*
> | Model | Computation Ratio | Memory Ratio |
> |------------|---------------------------|--------------------------|
> | Similarity-Inform SMoE | 1.048 | 1.008 |
> | Attention-Inform SMoE | 1.070 | 1.060 |
> | Similarity-Inform XMoE | 1.026 | 1.009 |
> | Attention-Inform XMoE | 1.038 | 1.060 |
> | Similarity-Inform SMoE-dropout | 1.047 | 1.008 |
> | Attention-Inform SMoE-dropout | 1.064 | 1.060 |

---

> ### Author Response · Authors · 2024-11-20
> **Review reply (2/3): Q3, Q4, Q5, Q6, Q7**
>
> **Q3 The introduction is structured unconventionally and I found myself taking time to find the motivation for the work.**
>
> **Reply:** Thank you for your comments. We will include the discussion about the motivation of the work with reviewers into the introduction of the paper.
>
> **Q4 It would be helpful to understand how the performance of M-I SMoE changes as the number of experts is increased, the granularity  of the experts changes, and the number of active parameters changes.**
>
> **Reply:** Following your suggestion, we evaluated our Mutual-Inform SMoE and baseline SMoE across different model configurations. While our original submission included experiments with 16 experts and varying active experts (top-1, top-2), we have now added new experiments with 32 experts and top-2, as well as 16 experts and top-8 routing during the rebuttal period to provide a more comprehensive analysis of trends. Across these settings, both Similarity-inform SMoE and Attention-inform SMoE consistently demonstrate better performance compared to the baseline SMoE, achieving lower PPL scores on both clean and attacked Wikitext-103 datasets (Table 3 below, or Table 6 in Appendix D.6 of our revision). When using 32 experts, our models achieve PPL reductions of up to 1.56 PPL compared to the corresponding baseline. When increasing to top-8 active experts, our models maintain their advantage with improvements of up to 1.64 PPL. These consistent performance gains across different architectural configurations further corroborate the robustness and effectiveness of our proposed models.
>
> We are not clear about what the reviewer mean by the "granularity of experts". We would really appreaciate it if the reviewer could elaborate this point more.
>
> Table 3: *PPL evaluation (lower is better) with the clean and attacked Wikitext-103 test set of baseline SMoEs and Mutual-Inform SMoE(s) with different number of experts and Top-M.*
> | Model/Metric | Clean Wikitext-103 Valid PPL | Clean Wikitext-103 Test PPL | Attacked Wikitext-103 Valid PPL | Attacked Wikitext-103 Test PPL |
> |------------|---------------------------|--------------------------|--------------------------------|-------------------------------|
> | SMoE (M = 1, K = 16) | 39.55 | 40.75 | 48.82 | 50.21 |
> | Similarity-inform SMoE (M = 1, K = 16) | 37.78 | 39.18 | 46.93 | 48.66 |
> | Attention-inform SMoE (M = 1, K = 16) | 38.02 | 39.35 | 47.20 | 48.72 |
> | SMoE (M = 2, K = 16) | 33.29 | 34.84 | 41.75 | 43.59 |
> | Similarity-inform SMoE (M = 2, K = 16) | **30.75** | **32.03** | **38.33** | **39.92** |
> | Attention-inform SMoE (M = 2, K = 16) | 31.31 | 32.23 | 39.68 | 40.91 |
> | SMoE (M = 8, K = 16) | 33.48 | 34.92 | 41.36 | 42.98 |
> | Similarity-inform SMoE (M = 8, K = 16) | 32.5 | 33.81 | 40.6 | 42.37 |
> | Attention-inform SMoE (M = 8, K = 16) | 31.97 | **33.28** | 39.98 | **41.45** |
> | SMoE (M=2, K = 32) | 31.82 | 33.41 | **39.90** | 41.79 |
> | Similarity-inform SMoE (M=2, K = 32) | 30.41 | **31.62** | 38.23 | 39.77 |
> | Attention-inform SMoE (M=2, K = 32) | **30.39** | 31.85 | **37.8** | **39.65** |
>
>
> **Q5 The architecture of the MoEs used (e.g., number of experts per layer) should be reported in the main text, not only in the appendix.**
>
> **Reply:** Thank you for your suggestion. We moved the architecture description of the models to the main text in our revised manuscript.
>
> **Q6 I am unable to find the dimensions of the transformer used in the language modeling experiments.**
>
> **Reply:** Thank you for pointing this out. Table 4 below summarizes the transformer model dimensions, number of layers and number of parameters. We have also added These architectural specifications in Appendix C1 and C2 of our revised manuscript.
> Table 4: Summarization of baselines' sizes
> | Baselines size | Number of params | Num layers | Model size |
> |------------|---------------------------|--------------------------|--------------------------------|
> | VMoE | 60M | 8 | 512 |
> | SMoE-medium | 215M | 6 | 352 |
> | SMoE-large | 388M | 12 | 512 |
> | GLAM-medium | 201M | 4 | 352 |
>
> **Q7 "Motivating by this PGM framework" Motivating --> Motivated
> Page 10: "We leave it for future work. We leave it for future work"**
>
> **Reply:** Thank you for pointing out these typos. We fixed all of them in the revision of the paper.

---

> ### Author Response · Authors · 2024-11-20
> **Review reply (3/3): Q8**
>
> **Q8 Is routing fluctuation discussed as a problem for MoEs outside of papers [1,2]?**
>
> **Reply:** Thank you for your question. In our revision (Section 6), we have added two references that confirm the existence of the routing fluctuation phenomenon in the two concurrent works [1, 2]. [1] mentions that various SMoE routers [3, 4] suffer from routing fluctuation without proposing solutions. Meanwhile [2] suggests that due to the variation of learnable parameters in the router. Therefore, we have also added [5] which provides another solution to improve the stability of the model. The work in [5] initially randomizes and freezes the router during training to provide a more stable routing strategy. Compared to those works, our paper aims to provide both a theoretical understanding of the attention-MoE and its current limitation, which includes routing fluctuation and non-robustness, and offers practical solutions for these limitations.
>
> **References**
>
> [1]Nguyen et al. "*LIBMoE: A Library for comprehensive benchmarking Mixture of Experts in Large Language Models.*" 2024.
>
> [2]Su et al. "*MaskMoE: Boosting Token-Level Learning via Routing Mask in Mixture-of-Experts*". 2024.
>
> [3]Csordas et al. "*Approximating two-layer feedforward networks for efficient transformers*." 2023.
>
> [4]Do et al."*Hyperrouter: Towards efficient training and inference of sparse mixture of experts*". ACL. 2023
>
> [5]Chen et al. "*Sparse MoE as the New Dropout: Scaling Dense and Self-Slimmable Transformers*" ICLR. 2023.
>
> -----
> We hope we have cleared your concerns about our work. We have also revised our manuscript according to your comments, and we would appreciate it if we can get your further feedback at your earliest convenience.

---

> ### Author Response · Authors · 2024-11-22
> **Any Questions from Reviewer jpxc on Our Rebuttal?**
>
> We would like to thank the reviewer again for your thoughtful reviews and valuable feedback.
>
> We would appreciate it if you could let us know if our responses have addressed your concerns and whether you still have any other questions about our rebuttal.
>
> We would be happy to do any follow-up discussion or address any additional comments.

---

### Official Review · Reviewer_rSWG · 2024-11-09

**Soundness:** 3
**Presentation:** 2
**Contribution:** 2
**Rating:** 5
**Confidence:** 3

**Summary:**

This paper formalized the transformer attention head as a probabilistic graph model and suggested that independent expert selections for different tokens in Sparse MoE contributes to the routing fluctuations. Based on such analyses, the authors proposed two intuitive approaches of aggregating other tokens' expert selections into the decision of the current token with either similarity between output features or adapted attention scores as the weights. Experiments in the paper demonstrated that such techniques could bring the improvements in terms of model performance and routing stability.

**Strengths:**

The structure of the paper is well-organized and comprehensive. The paper clearly identified the potential issues behind routing fluctuations at start and formed two solutions step by step.

**Weaknesses:**

1. The paper's writing did not fully meet the standard of a professional research publication .
- Lack of rigorous and concise formula definitions although the authors tried to analyze the problems in a formal way.
**a**. Diverse meanings of the same subscript in different formula. In Line 75-77, the subscript should be $k$ rather than $i$. It is not appropriate to use $i$ in line 87. Please do not confuse these two index symbols and try to keep one index have a unique meaning
**b**. Some notations lack explanations. What does the $f$ in Eq. (3) mean? What does $\hat{E}$ in Figure 1 mean? In Eq. (1), A and V should have subscripts. Also, the distribution whose mean is calculated should be listed under $E$ in Eq. (5) & (8).
**c**. Some equations have errors. In line 87-93, It would be better if the renormalization operation is taken over M rather than N. Namely, the denominator in Line should be the sum of M items since the remaining (K-M) scores are $-\infty$. In Eq. (4), "| X=X".
**d**. For some conventional definitions like the meaning of P(A) and E(A) in line 152-153, it is not necessary to point out them in the main text.
**e**. Putting lots of formulas/notations into the paper sometimes makes the content hard to follow if there is not enough illustrations and explanations. For example, Figure 1 is hard to parse without any captions about the definitions of symbols. It could be companied with graph illustrations of attention blocks or Sparse MoE.
- Redundant text. The preliminary and approach sections are redundant.  In Sec 1.2., there is no need to put too many words to explain the commonly-known fact that selecting top M values to calculate softmax is equivalent to the normalization over top M values after softmax. and I am not convinced that this explanation is necessary for further analyzes.  The explanations of two context-dependent expert selection methods could be condensed. The paper should be delivered in a concise way with compact information.
- Some typos. In line 20,  "Motivated by" not "Motivating by". In line 166, it should be "Considering". In line 255, 261, 267, there are some indexes (5, 6, 7) coming from nowhere. In line 290, it should be "similar to". In 128-129 “is as” and a misplaced comma seems confusing to me.
- Others. The EMPIRICAL ANALYSIS and EXPERIMENTAL RESULTS could be merged rather than set apart.


2. The novelty of the paper is questionable. There is lack of sufficient literature review. The papers listed in the related sections seem to be "out-of-date", i.e. published two years ago and the proposed methods were compared against a limited set of long time-established baselines. Although it is interesting to see that the authors fit the MoE into PGM framework, the proposed "attention-like" aggregation techniques are still quite simple and straightforward, making me question the necessity of using this complicated PGM framework.

3. Experiments. In Figure 2, the entropy curve for the baseline is missing in the right plot, and the dynamics of fluctuation values across different layers should be analyzed.

**Questions:**

See the above weaknesses part.

---

> ### Author Response · Authors · 2024-11-20
> **Review Reply (1/5): Q1**
>
> **Q1.1 Lack of rigorous and concise formula definitions although the authors tried to analyze the problems in a formal way.**
>
> **Reply:** Thanks for your comment. Although the formula definitions in our paper could be made more concise, we respectfully disagree with the reviewer’s assertion that they lack rigor. We would like to clarify about our comprehensive mathematical framework throughout its development. The foundational probabilistic graphical models for MoE-Transformer are formally defined in Sections 2.1-2.2. To interpret the Attention-MoE block as a point estimate of a three-layer hierarchical mixture of expert regression, we provide explicit generating processes and detailed conditional probability specifications for each the corresponding graph $\mathcal{G}_1$ and derived the result, step by step from establishing optimal regression functions for inference on the probabilistic model (Equations 4) to the derivation of attention mechanisms (end of section 2.1) and finally, Attention-MoE block as a point estimate of the optimal regression function for the PGM model $\mathcal{G}_1$. Similarly, in section 3.1 and 3.2, we derives the novel Mutual-Inform (S)MoE (including Similarity-Inform (S)MoE and Attention-Inform (S)MoE) in the same principled way based on the graph $\mathcal{G}_2$ and $\mathcal{G}_3$, respectively, with Similarity-Inform SMoE is defined in Definition 1 (Section 3.1) and Attention-Inform SMoE is defined in in Definition 2 (Section 3.2). In Proposition 2 (Section 3.3), we also theoretically prove that Mutual-Inform (S)MoEs reduce routing fluctuation by reducing the entropy of token decision.
>
> Complete mathematical proofs and detailed derivations are provided in the appendices to ensure reproducibility and verify the mathematical soundness of our approach
>
> **Q1.2 Diverse meanings of the same subscript in different formula. In Line 75-77, the subscript should be $k$ rather than $i$. It is not appropriate to use $i$ in line 87. Please do not confuse these two index symbols and try to keep one index have a unique meaning b. Some notations lack explanations. What does the $f$ in Eq. (3) mean? What does $\tilde{E}$ in Figure 1 mean? In Eq. (1), A and V should have subscripts. Also, the distribution whose mean is calculated should be listed underin Eq. (5) & (8). c. Some equations have errors. In line 87-93, It would be better if the renormalization operation is taken over M rather than N. Namely, the denominator in Line should be the sum of M items since the remaining (K-M) scores are. In Eq. (4), "| X=X". d. Conventional definitions like the meaning of P(A) and E(A) in line 152-153, it is not necessary to point out them in the main text. Putting lots of formulas/notations into the paper makes the content hard to follow if there is not enough illustrations and explanations. For example, Figure 1 is hard to parse without any captions about the definitions of symbols. It could be companied with graph illustrations of attention blocks or Sparse MoE.**
>
> **Reply:** Thank you for pointing out typos in the paper. We fixed all mentioned typos in the revison of our paper in blue color.   We would like to explain the mentioned notations as follow. The $f$ in Eq. (3) is actually a typo, we changed it to $g_k$ i.e., the $k$-th expert function. The $\tilde{E}$ in Figure 1 is the stacked of $[{\tilde{e}_1,\ldots,\tilde{e}_N}]$. We will move the definition of P(A) and E(A) to Appendix as suggested by the reviewer. We revised the Figure 1 by explaining notations inside the figure in its caption. While we agree with the reviewer that the paper contains some typos, we still believe that the problems are analyzed in a formal way.

---

> ### Author Response · Authors · 2024-11-20
> **Review Reply (2/5): Q1, Q3**
>
> **Q1.3 Redundant text. The preliminary and approach sections are redundant. In Sec 1.2., there is no need to put too many words to explain the commonly-known fact that selecting top M values to calculate softmax is equivalent to the normalization over top M values after softmax. and I am not convinced that this explanation is necessary for further analyzes. The explanations of two context-dependent expert selection methods could be condensed. The paper should be delivered in a concise way with compact information.**
>
> **Reply:** Thank you for your constructive feedback. We will move the discussion on normalization and expert selection to the Appendix.
>
> **Q1.4 Some typos. In line 20, "Motivated by" not "Motivating by". In line 166, it should be "Considering". In line 255, 261, 267, there are some indexes (5, 6, 7) coming from nowhere. In line 290, it should be "similar to". In 128-129 “is as” and a misplaced comma seems confusing to me.**
>
> **Reply:** Thank you for pointing out the typos. We fixed all of them in the revision of the paper. The indexes (5, 6, 7) is used to continue the generation process (1,2,3) in Section 2.1 and (4) in Section 2.2 since the proposed MoEs shares some components from the conventional multihead Attention and MoEs. We added an explanation in the revision.
>
> **Q1.5 Others. The EMPIRICAL ANALYSIS and EXPERIMENTAL RESULTS could be merged rather than set apart.**
>
> **Reply:** Thank you for your suggestion. We will merge the two sections in our revision.
>
> **Q3 Experiments. In Figure 2, the entropy curve for the baseline is missing in the right plot, and the dynamics of fluctuation values across different layers should be analyzed.**
>
> **Reply:** In Figure 2's right plot, we represent entropy ratios relative to the SMoE baseline, where the curve for the baseline would be a constant line at $y = 1$. We chose to omit this line to maintain visual clarity.

---

> ### Author Response · Authors · 2024-11-20
> **Review Reply (3/5): Q2**
>
> **Q2.1There is lack of sufficient literature review. The papers listed in the related sections seem to be "out-of-date", i.e. published two years ago and the proposed methods were compared against a limited set of long time-established baselines.**
>
> **Reply**: We agree with the reviewer that some recent papers on mitigating routing fluctuation in SMoE are missing in our related work discussion. In our revision (Section 6), we have added two references that confirm the existence of the routing fluctuation phenomenon in the two concurrent works [1, 2]. [1] mentions that various SMoE routers [3, 4] suffer from routing fluctuation without proposing solutions. Meanwhile, [2] suggests that due to the variation of learnable parameters in the router. We have also added [5] which provides another solution to improve the stability of the model ( in Section 6 of our revision). The work in [5] initially randomizes and freezes the router during training to provide a stable routing strategy. Compared to those works, our paper aims to provide both a theoretical understanding of the attention-MoE and its current limitation, which includes not only routing fluctuation but also non-robustness, and offer practical solutions for these limitations.
>
> In addition, we have conducted experiments comparing our Mutual-Inform SMoE models with X-MoE [6], previous work that addresses routing fluctuation in MoE models, and the more recent SMoE-dropout[5]. While both X-MoE and SMoE-dropout show improved performance over the standard SMoE baseline, as shown in Table 1 below (or Table 3, Appendix D.3), our Mutual-inform SMoE variants still significantly outperform them, as evidenced by the lower PPL scores across both clean and attacked Wikitext-103 datasets. Furthermore, when we integrate our proposed methods with X-MoE and SMoE-dropout to create Similarity-inform X-MoE, Attention-inform X-MoE, Similarity-inform SMoE-dropout, and Attention-inform SMoE-dropout variants, we observe substantial improvements over their respective baselines, with consistent PPL reductions in both validation and test sets. These results demonstrate not only the advantages of our approach but also its effectiveness as a plug-and-play solution that can enhance various MoE architectures.
>
> Table 1: *PPL evaluation (lower is better) with the clean and attacked Wikitext-103 on valid set and test set of SMoE-medium size variants, M=2*
> | Model/Metric | Clean Wikitext-103 Valid PPL | Clean Wikitext-103 Test PPL | Attacked Wikitext-103 Valid PPL | Attacked Wikitext-103 Test PPL |
> |------------|---------------------------|--------------------------|--------------------------------|-------------------------------|
> | SMoE | 33.29 | 34.84 | 41.75 | 43.59 |
> | Similarity-inform SMoE | **30.75** | **32.03** | **38.33** | **39.92** |
> | Attention-inform SMoE | 31.31 | 32.23 | 39.68 | 40.91 |
> | X-MoE | 33.05 | 34.49 | 41.68 | 42.96 |
> | Similarity-inform X-MoE | **31.83** | **33.06** | **39.92** | **41.28** |
> | Attention-inform X-MoE | 32.06 | 33.24 | 40.35 | 41.73 |
> | SMoE-dropout | 33.08 | 34.67 | 41.11 | 43.09 |
> | Similarity-inform SMoE-dropout | 32.47 | 33.69 | 40.6 | 41.99 |
> | Attention-inform SMoE-dropout | **32.21** | **33.91** | **40.56** | **42.17** |
>
> **References**
>
> [1]Nguyen, et al. "*LIBMoE: A Library for comprehensive benchmarking Mixture of Experts in Large Language Models.*" 2024.
>
> [2]Su, et al. "*MaskMoE: Boosting Token-Level Learning via Routing Mask in Mixture-of-Experts*". 2024.
>
> [3]Csordas, et al. "*Approximating two-layer feedforward networks for efficient transformers*." 2023.
>
> [4]Do, et al."*Hyperrouter: Towards efficient training and inference of sparse mixture of experts*". ACL. 2023
>
> [5]Chen, et al. "*Sparse MoE as the New Dropout: Scaling Dense and Self-Slimmable Transformers*" ICLR. 2023.
>
> [6]Chi et al. "On the Representation Collapse of Sparse Mixture of Experts". NeurIPS. 2022.

---

> > ### Author Response · Authors · 2024-11-20
> > **Review Reply (4/5): Q2**
> >
> > **Q2.2 Although it is interesting to see that the authors fit the MoE into PGM framework, the proposed "attention-like" aggregation techniques are still quite simple and straightforward, making me question the necessity of using this complicated PGM framework.**
> >
> > **Reply**: Thank you very much for your comments. Being simple and straightforward is not a weakness but the advantage of a new method. In fact, our criteria for developing a new machine learning method is: 1) the method is simple to understand and straightforward to implement, 2) the method really works on practical tasks, and 3) the method is derived from first principles that provide us tools and techniques for further extension and for deriving deep theoretical understandings of the method. Our proposed Mutual-Inform SMoE (Attention-Inform SMoE and Similarity-Inform SMoE) aggregation satisfy all of these criteria. First, Mutual-Inform SMoE such as Attention-Inform is simple and straightforward as the reviewer pointed out. Second, empirical results in Tables 1, 2, 3, 4, 5, 6, 7 and Figures 2, 3, 4 in our revised manuscript justify the advantages of our proposed Mutual-Inform SMoEs aggregation over the baseline SMoE on a variety of practical tasks. Finally, the proposed Mutual-Inform SMoE aggregation results from our PGM framework, which provides a foundational tool to further design our Attention-Inform and Similarity-Inform (S)MoE into new families of Attention-(S)MoE models. This PGM framework also allows us to prove the stability of Attention-Inform and Similarity-Inform (S)MoEs in Proposition 1 in our manuscript and offer a better understanding of Attention-(S)MoE models.
> >
> > On the last point above, please allow us to take this chance to clarify the necessity of using our PGM framework. In particular, the benefits of the PGM framework is two-fold:
> >
> > * **Our PGM framework provides a principled understanding of Attention-(S)MoE mechanism.** The framework provides a formal mathematical foundation for understanding attention-MoE block. It reveals why token routing decisions are conditionally independent given attention input $X$ or (S)MoE input $U$ (see Section 2.2 of our manuscript). Without viewing the entire transformer (attention-MoE) block as a PGM with token decisions as random variables, it would be more difficult to identify why these variables are mutually conditionally independent. This insight helps explain the cause of routing fluctuation (Section 2.2) and suggests pathways for improvement (Section 3).
> >
> > * **Our PGM framework provides a principled way to design Attention-(S)MoE**. The PGM framework not only aids understanding but provides a principled approach to designing Attention-SMoE. In this work, we investigate the mutual-inform design of (S)MoE router to improve the accuracy and robustness of the model. Our Mutual-Inform (S)MoE introduces a final decision variable for each token $\tilde{d}_i$, which is directly depends on tokens' decision before receiving decision information from other tokens, $\tilde{e}_j$ for $j = 1, \dots, N$, and correpondance between tokens. In the case of Similarity-Inform (S)MoE, the correspondance score is captured in the similarity variable $\tilde{s}_i$ of $U_i$, thus its computation is fairly straitforward, as in Eqn.(6) in Section. 3.1.
> >
> >      However, in the case that the correspondance is captured in the attention, as in Attention-Inform (S)MoE, the proposed model is not trivial to derive without the PGM framework. In this method, we establish a link between variable $\tilde{z}_i$ (the attention variable defined in Sec 2.1, line 181, 182) and token's final routing decision $\tilde{d}_i$. To study the effect of this link in final decision, we need to relies on the conditional independencies captured by the PGM model. The resulting derivation of Attention-inform SMoE in Section 3.2 can be interpreted as the following two-stage influence process.
> >     * First, each token's original decision is adjusted by the decisions of other tokens, weighted by $A'_h[i,j]$ the posterior distribution of attention variable given the observation of MoE input $u_i$ (as well as the head index and input X). This represents the "responsibility" of token $j$ in explaining token $i$'s representation within the attention head.
> >     * Then, these weighted decisions from each head are further weightedly combined by $H'[i,h]$, the posterior probability of head index given input $u_i$ of the (S)MoE and input $X$ of the attention. This represents the responsibility of head $h$ in explaining token $i$.
> >
> >     Underlying insight of the seemingly simple link between $\tilde{z}_i$ and $\tilde{d}_i$ results in hierarchical weighting scheme, allowing the model to integrate context from multiple attention patterns, which is very difficult to derived without carefully investigating the proposed PGM framework.

---

> ### Author Response · Authors · 2024-11-20
> **Review Reply (5/5): Q2**
>
> **Q2.3 The novelty of the paper is questionable.**
>
> **Reply:** Thank you for your comment. We respectfully disagree with the reviewer’s comment that the novelty of our paper is questionable. Please allow us to clarify our key contributions and novelties below.
>
> First, to the best of our knowledge, our paper is the first work that develops a probabilistic graphical models (PGM) for MoE Transformer. In particular, we present a novel probabilistic graphical framework (PGM) and then derive the exact architecture of MoE Transformer as a point estimate of the regression function in a three-layer hierarchical mixture of experts. While there have been other works that try to derive a either transformer/attention layer or a MoE layer from different perspectives such as using k-SVD [1], nonlocal variational denoising [2], or Markov Chain [3], we are the first to ***derive the full 2-layer module of a transformer layer following by an MoE layer*** using PGM. Compared to previous work, our results are not limited to one layer only and were derived from a totally different approach, which allows us to understand and manipulate the conditional independencies of the underlying graph to obtain the corresponding MoE-Transformer mechanism.
>
> Second, our paper is again the first work that ***explores the use of token similarity to inform the routing in MoE***. The motivation behind this idea is that a token's routing decision should directly influence other tokens' routing decisions. Our innovative approach yields the following benefits that we demonstrate both theoretically and empirically via our proposed Attention-Inform and Similarity-Inform (S)MoE:
>  * Reduced routing fluctuation and enhanced model robustness (Section 3.3 and Section 4, D)
>  * Improved load balancing (Section D.2)
>  * Improve model performance (Section 4, D, E)
>
> **Extensible Framework**: The mutual-inform concept we introduce extends beyond just similarity and attention-based mechanisms. This opens new research directions for exploring various token-informed routing strategies, which we identify as our future work.
>
> **References**
>
> [1]Chen et al. "*Primal-Attention: Self-attention through Asymmetric Kernel SVD in Primal Representation*". NeurIPS. 2023.
>
> [2]Nguyen et al. "*Mitigating Over-smoothing in Transformers via Regularized Nonlocal Functionals*". NeurIPS. 2024.
>
> [3]Ildiz et al. "*From Self-Attention to Markov Models: Unveiling the Dynamics of Generative Transformers*". ICML. 2024.
>
> -----
> We hope we have cleared your concerns about our work. We have also revised our manuscript according to your comments, and we would appreciate it if we can get your further feedback at your earliest convenience.

---

> > ### Author Response · Authors · 2024-11-22
> > **Any Questions from Reviewer rSWG on Our Rebuttal?**
> >
> > We would like to thank the reviewer again for your thoughtful reviews and valuable feedback.
> >
> > We would appreciate it if you could let us know if our responses have addressed your concerns and whether you still have any other questions about our rebuttal.
> >
> > We would be happy to do any follow-up discussion or address any additional comments.

---

### Author Response · Authors · 2024-11-20
**Summary of revision**

Incorporating the insightful comments from the reviewers into our paper has helped us improve its quality. Here, we provide a summary of the revisions (highlighted in blue in the revised manuscript) for convenience:
1. We fixed typos suggested by reviewers.
2. Section 6: we added the discussion of more recent related works.
3. Appendix D.3: we compare and combine Mutual-Inform SMoEs with previous works to confirm the benefit of our methods.
4. Appendix D.4: we evaluate the adaptability of our models, we examine the fine-tuning performance of pretrained SMoE variants on different downstream tasks.
5. Appendix D.5: we demonstrate the scalability of our models by evaluating them with a larger transformer-MoEs baseline.
6. Appendix D.6: we evaluate performance across different model configurations, we experiment with varying numbers of experts and active experts.
7. Appendix D.7: we compare the computational complexity and memory complexity of using mutual inform techniques to the baselines.
8. Appendix C.1 and C.2: we add the model size for the baselines.

---

### Author Response · Authors · 2024-11-20
**General response (2/3)**

2.**Adaptability of our models**. To evaluate the adaptability of our models, we examine the fine-tuning performance of pretrained SMoE variants on different downstream tasks. Specifically, we assess their test accuracy on Stanford Sentiment Treebank 5[1], 2[2] (SST5, SST2), and Banking-77[3] (B77) datasets, as shown in Table 2 (or see Table 4, Appendix D.4). The results demonstrate that Attention-inform SMoE consistently achieves the highest accuracy across all datasets (38.89% on SST5, 72.41% on SST2, and 85.84% on B77), while Similarity-inform SMoE also consistently outperforms the conventional SMoE baseline. These improvements in fine-tuning performance across diverse tasks suggest that our proposed Mutual-inform SMoE variants possess better adaptability compared to conventional SMoEs.

Table 2: *Top-1 test accuracy on Stanford Sentiment Treebank 5, 2 (SST5, SST2), and Banking-77 (B77) finetuning task for SMoE-medium size variants, M=2*
| Model | sst5 | sst2 | banking77 |
|------------|---------------------------|--------------------------|--------------------------------|
| SMoE | 36.54 | 70.23 | 83.96 |
| Similarity-inform SMoE | 37.91 | 71.72 | 85.19 |
| Attention-inform SMoE | **38.89** | **72.41** | **85.84** |

**References**

[1, 2]Socher et al. "*Recursive Deep Models for Semantic Compositionality Over a Sentiment Treebank.*" EMNLP. 2013.

[3]Casanueva et al. "*Efficient intent detection with dual sentence encoders.*" NLP4ConvAI. 2013.

3.**Scalability of our models** To further demonstrate the scalability of our models, we evaluate them with a larger transformer-MoEs baseline of approximately 390M parameters, with 12 layers. The results in Table 3 (or Table 5 in Appendix D.5 ) confirms that the scaling law holds true, as all models show improved language modeling performance with increased parameter count. Importantly, both Similarity-inform SMoE and Attention-inform SMoE maintain their performance advantage over the conventional SMoE at this larger scale, with Similarity-inform SMoE emerging as the best performing variant. These findings validate that the benefits of our proposed methods are preserved when scaling up model size.

Table 3: *PPL evaluation with the clean and attacked Wikitext-103 of ***SMoE-large*** size variants, M=2*
| Model/Metric | Clean Wikitext-103 Valid PPL | Clean Wikitext-103 Test PPL | Attacked Wikitext-103 Valid PPL | Attacked Wikitext-103 Test PPL |
|------------|---------------------------|--------------------------|--------------------------------|-------------------------------|
| SMoE | 28.737 | 30.378 | 36.43 | 38.34 |
| Similarity-inform SMoE | **27.06** | **28.34** | **34.65** | **36.28** |
| Attention-inform SMoE | 27.26 | 28.69 | 34.69 | 36.37 |

4.**Different number of experts and Top-M**. While our original submission included experiments with 16 experts and varying active experts (top-1, top-2), we have now added new experiments with 32 experts and top-2, as well as 16 experts and top-8 routing during the rebuttal period to provide a more comprehensive analysis of trends. Across all these settings, both Similarity-inform SMoE and Attention-inform SMoE perform consistently better than the baseline SMoE, achieving lower PPL scores on both Clean and Attacked Wikitext-103 datasets (Table 4, or Table 6 in Appendix D.6). When using 32 experts, our methods achieve PPL reductions of up to 1.56 PPL compared to the baseline, and when increasing to top-8 active experts, they maintain their advantage with improvements of up to 1.64 PPL. These performance gains further demonstrate the robustness and effectiveness of our models.

Table 4: *PPL evaluation (lower is better) with the clean and attacked Wikitext-103 of baseline SMoEs and Mutual-Inform SMoE(s) with different number of experts and Top-M.*
| Model/Metric | Clean Valid PPL | Clean Test PPL | Attacked Valid PPL | Attacked Test PPL |
|------------|---------------------------|--------------------------|--------------------------------|-------------------------------|
| SMoE (M = 1, K = 16) | 39.55 | 40.75 | 48.82 | 50.21 |
| Similarity-inform SMoE (M = 1, K = 16) | 37.78 | 39.18 | 46.93 | 48.66 |
| Attention-inform SMoE (M = 1, K = 16) | 38.02 | 39.35 | 47.20 | 48.72 |
| SMoE (M = 2, K = 16) | 33.29 | 34.84 | 41.75 | 43.59 |
| Similarity-inform SMoE (M = 2, K = 16) | **30.75** | **32.03** | **38.33** | **39.92** |
| Attention-inform SMoE (M = 2, K = 16) | 31.31 | 32.23 | 39.68 | 40.91 |
| SMoE (M = 8, K = 16) | 33.48 | 34.92 | 41.36 | 42.98 |
| Similarity-inform SMoE (M = 8, K = 16) | 32.5 | 33.81 | 40.6 | 42.37 |
| Attention-inform SMoE (M = 8, K = 16) | 31.97 | **33.28** | 39.98 | **41.45** |
| SMoE (M=2, K = 32) | 31.82 | 33.41 | **39.90** | 41.79 |
| Similarity-inform SMoE (M=2, K = 32) | 30.41 | **31.62** | 38.23 | 39.77 |
| Attention-inform SMoE (M=2, K = 32) | **30.39** | 31.85 | **37.8** | **39.65** |

---

> ### Author Response · Authors · 2024-11-20
> **General response (3/3)**
>
> We would also like to summarize our work's key contributions below.
>
> Our work develops a probabilistic graphical models (PGM) for MoE Transformer. In particular, we present a novel probabilistic graphical framework (PGM) and then derive the exact architecture of MoE Transformer as a point estimate of the regression function in a three-layer hierarchical mixture of experts. While there have been other works that try to derive a either transformer/attention layer or a MoE layer from different perspectives such as using k-SVD [1], nonlocal variational denoising [2], or Markov Chain [3], , to our best knowledge, we are the first to ***derive the full 2-layer module of a transformer layer following by an MoE layer*** using PGM. Compared to previous work, our results are not limited to one layer only and were derived from a totally different approach, provide us a principled way to understand and manipulate the conditional independencies of the underlying graph to obtain the corresponding MoE-Transformer mechanism.
>
> Second, our paper ***explores the use of token similarity to inform the routing in MoE***. The motivation behind this idea is that a token's routing decision should directly influence other tokens' routing decisions. Our approach yields the following benefits that we demonstrate both theoretically and empirically via our proposed Attention-Inform and Similarity-Inform (S)MoE:
>  * Reduced routing fluctuation and enhanced model robustness (Section 3.3 and Section 4, D)
>  * Improved load balancing (Section D.2)
>  * Improve model performance (Section 4, D, E)
>
> **References**
>
> [1]Chen et al. "*Primal-Attention: Self-attention through Asymmetric Kernel SVD in Primal Representation*". NeurIPS. 2023.
>
> [2]Nguyen et al. "*Mitigating Over-smoothing in Transformers via Regularized Nonlocal Functionals*". NeurIPS. 2024.
>
> [3]Ildiz et al. "*From Self-Attention to Markov Models: Unveiling the Dynamics of Generative Transformers*". ICML. 2024
>
> -----
>
> We are glad to answer any further questions you have on our submission.

---

### Author Response · Authors · 2024-11-20
**General Response (1/2)**

Dear AC and Reviewers,

Thanks for your thoughtful reviews and valuable comments, which have helped us significantly improve the paper. We are encouraged by your endorsements that: 1) Our paper is interesting and provide valuable insights (Reviewer jpxc), its strength lies in its novelty (Reviewer jpxc, Reviewer Mg8s, Reviewer azAZ), also Reviewer azAZ highlights our novelty in innovatively reinterpreting SMoE which is a fresh way of addressing the critical issue of routing challenges; (2) our work is theoretically solid (Reviewer jpxc, Reviewer Mg8s, Reviewer azAZ), with sound, rigorous and well-presented mathematical derivations and proofs backing claims (Reviewer azAZ, Reviewer Mg8s); (3) our method and frameworkare are elegant, practical, comprehensive and rigorous (Reviewer Mg8s).
Reviewer azAZ also states that “improvements in both standard and adversarial settings underscore the paper’s value in robust model deployment”. We have also updated our submission based on the reviewers' feedbacks, and we have highlighted our revision in blue.

One of the main requests from reviewers is to extend our baselines to previous works and include more diverse baseline approaches. The reviewers also raised questions about the adaptability of our Mutual-inform SMoE and its performance across different settings, such as increases in the number of experts, number of active parameters, and the overall scalability of the model. We address these questions here.

1.**Comparisons to previous works**. To further investigate the advantages of our Mutual-inform SMoE, we compare and adapt our proposed models to X-MoE, previous work that addresses routing fluctuation in MoE models, and SMoE-dropout, another method that improves upon the standard SMoE. While both X-MoE[1] and SMoE-dropout[2] show improved performance over the standard SMoE baseline, as shown in Table 1 (or Table 3, Appendix D.3), our Mutual-inform SMoE variants still significantly outperform them, as evidenced by the lower PPL scores across both Clean and Attacked Wikitext-103 datasets. Furthermore, when we integrate our proposed methods with these models to create Similarity-inform X-MoE, Attention-inform X-MoE, Similarity-inform SMoE-dropout, and Attention-inform SMoE-dropout variants, we observe substantial improvements over their respective baselines, with consistent PPL reductions in both validation and test sets. These results demonstrate not only the superior performance of our approach but also its effectiveness as a plug-and-play solution that can enhance various MoE architectures and improve model robustness.

Table 1: *PPL evaluation (lower is better) with the clean and attacked Wikitext-103 on valid set and test set of SMoE-medium size variants, M=2*
| Model/Metric | Clean Wikitext-103 Valid PPL | Clean Wikitext-103 Test PPL | Attacked Wikitext-103 Valid PPL | Attacked Wikitext-103 Test PPL |
|------------|---------------------------|--------------------------|--------------------------------|-------------------------------|
| SMoE | 33.29 | 34.84 | 41.75 | 43.59 |
| Similarity-inform SMoE | **30.75** | **32.03** | **38.33** | **39.92** |
| Attention-inform SMoE | 31.31 | 32.23 | 39.68 | 40.91 |
| X-MoE | 33.05 | 34.49 | 41.68 | 42.96 |
| Similarity-inform X-MoE | **31.83** | **33.06** | **39.92** | **41.28** |
| Attention-inform X-MoE | 32.06 | 33.24 | 40.35 | 41.73 |
| SMoE-dropout | 33.08 | 34.67 | 41.11 | 43.09 |
| Similarity-inform SMoE-dropout | 32.47 | 33.69 | 40.6 | 41.99 |
| Attention-inform SMoE-dropout | **32.21** | **33.91** | **40.56** | **42.17** |

**References**

[1]Chi et al. "*On the Representation Collapse of Sparse Mixture of Experts*". NeurIPS. 2022

[2]Chen, et al. "*Sparse MoE as the New Dropout: Scaling Dense and Self-Slimmable Transformers*" ICLR. 2023.

---

### Meta-Review · Area_Chair_T2Ba · 2024-12-16

**Metareview:**

This paper proposes a probabilistic graphical model (PGM) framework to better understand and stabilize routing decisions in Sparse Mixture of Experts (SMoE) architectures. By identifying that standard SMoE methods rely on conditional independence assumptions leading to routing fluctuations, the authors introduce Mutual-Inform SMoEs, which incorporate inter-token influences via similarity or attention. They provide proofs of reduced routing entropy and empirically show that their approach can improve robustness and reduce fluctuations on mid-scale vision and language modeling tasks. Strengths include a theoretical motivation, a novel PGM perspective on SMoEs, and initial empirical evidence that the proposed method enhances routing stability and performance. However, several weaknesses limit the paper’s impact. The experiments are conducted on relatively small-scale models and with limited training durations, making it difficult to confidently assess scalability to large-scale LLM scenarios where MoEs are most relevant. Additionally, initial versions lacked thorough baseline comparisons and crucial details, only partly remedied during the discussion phase. Given the importance of large-scale validation for MoEs would benefit significantly from more extensive experiments, deeper comparisons, and clearer documentation of hyperparameters and methodologies.

**Additional Comments On Reviewer Discussion:**

During the rebuttal, reviewers raised concerns about missing comparisons to recent methods, limited scaling experiments, insufficient training length, unclear proofs, and a lack of hyperparameter and computational cost analysis. In response, the authors added experiments comparing their approach to X-MoE and SMoE-dropout, introduced results on a larger model, clarified their proofs and writing. However, jpxc points out that the experiments are still seen as limited in terms of scale for a paper focusing on MoE’s.

---

### Decision · Program_Chairs · 2025-01-22

Reject